# Generative Modeling Reinvents Supervised Learning: Label Repurposing with Predictive Consistency Learning

**Yang Li** [1 2]  **Jiale Ma** [1 2]  **Yebin Yang** [1 2]  **Qitian Wu** [3]  **Hongyuan Zha** [4]  **Junchi Yan** [1 2]

## Abstract

Predicting labels directly from data has been the standard in label learning tasks, e.g., supervised learning, where models often prioritize feature compression and extraction from inputs under the assumption that label information is less complex. However, recent prediction tasks often face predicting complex labels, exacerbating the challenge of learning mappings from learned features to high-fidelity label representations. To this end, we draw inspiration from the consistency training concept in generative consistency models and propose predictive consistency learning (PCL), a novel learning paradigm that decomposes the full label information into a progressive learning procedure, mitigating the label capture challenge. Besides data inputs, PCL additionally receives input from noise-perturbed labels as an additional reference, pursuing predictive consistency across different noise levels. It simultaneously learns the relationship between latent features and a spectrum of label information, which enables progressive learning for complex predictions and allows multi-step inference analogous to gradual denoising, thereby enhancing the prediction quality. Experiments on vision, text, and graph tasks show the superiority of PCL over conventional supervised training in complex label prediction tasks.

## 1. Introduction

Machine learning has long relied on the paradigm of mapping input data to corresponding output labels by minimizing prediction error, where supervised learning is a prominent example. This direct label prediction paradigm has been widely applied across various domains, from image classification (Krizhevsky et al., 2012; He et al., 2016; Simonyan & Zisserman, 2014), natural language processing (Vaswani, 2017; Devlin, 2018; Radford, 2018), to structured graph learning (Kipf & Welling, 2016; Veličković et al., 2017; Wu et al., 2022) where it is considered a standard practice. In such systems, it is very typical to employ a neural network to directly map the data inputs to labels, with a particular focus on the expressive capacity of (deep) models to compress high-dimensional inputs into latent representations while preserving relevant information for accurate predictions, viewed from an information theory perspective (Tishby et al., 2000; Tishby & Zaslavsky, 2015). This compression is believed to contribute to the generalization ability of deep learning models, particularly in high-dimensional and noisy input scenarios.

This paradigm typically assumes that the labels contain a significantly lower dimensionality and less information than the data inputs, thus guiding model designs toward compressing and extracting relevant features from the input space for efficient prediction (Tishby & Zaslavsky, 2015). The assumption further implies that transforming meaningful latent features to label outputs is relatively straightforward compared to the challenge of extracting expressive features. However, recent advanced scenarios involve much more complex labels, leading to new challenges. Examples include image prediction extending to dense, pixel-level outputs (Long et al., 2015; Chen et al., 2017), natural language processing tasks generating complex sentences (Brown, 2020; Touvron et al., 2023), and predicting complex structured solutions based on graph representations (Li et al., 2023b; Satorras et al., 2021). These challenges expose predictive bottlenecks due to the inherent complexity of transformations from features to labels, in addition to feature extraction. To address this challenge, one approach involves learning an efficient representation of the complex labels, facilitating a more effective transformation within the low-dimensional feature space. Indeed, this can correspond to methods that leverage Variational Auto-Encoder (VAE) (Kingma, 2013) to perform learning tasks within the latent space (Rombach et al., 2022; Hottung et al., 2021). However, this approach necessitates that the transformation between labels and latent features

[1]School of Artificial Intelligence, Shanghai Jiao Tong University [2]Shanghai Innovation Institute [3]Broad Institute of MIT and Harvard [4]The Chinese University of Hong Kong, Shenzhen. Correspondence to: Junchi Yan <yanjunchi@sjtu.edu.cn>. This work is partly supported by NSFC 62222607. Code available at github repository.

*Proceedings of the 42nd International Conference on Machine Learning*, Vancouver, Canada. PMLR 267, 2025. Copyright 2025 by the author(s).

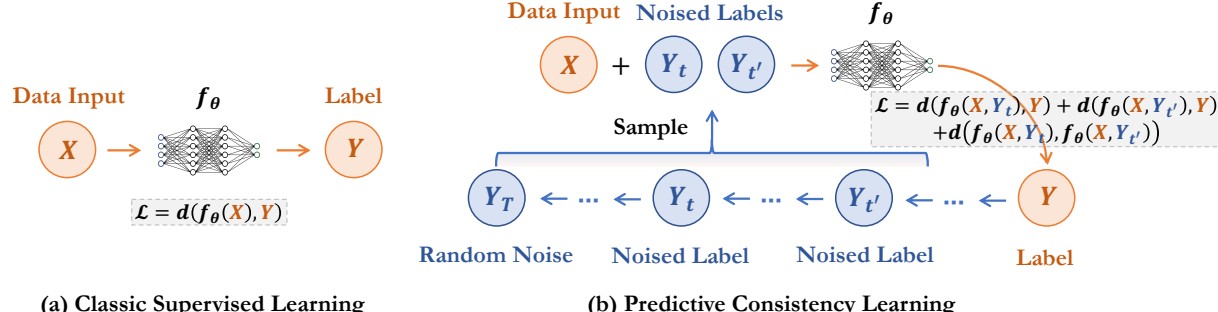

**(a) Classic Supervised Learning**  **(b) Predictive Consistency Learning**

*Figure 1.* Illustration of predictive consistency learning (PCL). Unlike traditional approaches that predict labels directly from inputs, PCL predicts labels using inputs and noise-perturbed label hints and pursues predictive consistency across different noise steps.

be reversible, requiring training additional neural networks.

In this paper, drawing inspiration from the concept of consistency mapping in generative consistency models (Song et al., 2023; Song & Dhariwal, 2024), we propose an alternative approach to better capture complex label information by introducing a fundamentally different learning paradigm. This learning scheme aims to decompose the full label information into a gradual process, where the model begins by simultaneously capturing partial label details and progressively approximates the complete label. To schedule the label information to be learned, we resort to the noising process of the diffusion and consistency models (Ho et al., 2020; Song et al., 2020; 2023) and generate noisy labels as additional input hints, enabling the model to learn the complementary information alongside the noisy part. The proposed Predictive Consistency Learning (PCL) frames the learning process as learning the predictive consistency from input label hints of various noise levels toward the target labels. This process can be viewed as a label generation process conditioned on the input data. However, to align with prediction scenarios, each training instance is paired with a reference target label, and the model learns to guide all denoising trajectories toward this target by enforcing predictive consistency across different noise timesteps, which we define as prediction consistency.

Specifically, during training, unlike conventional methods predicting labels directly from inputs, PCL maps noisy labels at varying noise levels back to the true label conditioned on the input data and enforces different noise timesteps mapping to the same target. By enforcing predictive consistency across multiple noise levels, it captures a rich spectrum of label information from entirely noisy to accurate predictions, fostering a more expressive mapping. During inference, the inherent multi-step denoising mechanism also facilitates progressive refinement, resulting in more flexible and accurate predictions. Intuitively, this process can be seen as learning to predict with varying degrees of solution hints, which benefits learning by progressively understanding the label in-

formation, especially when the labels are complex, while the prediction consistency serves as a mechanism to propagate prediction accuracy from low-noise to high-noise conditions and constrains the invariance of the learned representation across varying noise levels, promoting the information leverage from the input data. Fig. 1 illustrates the pipeline.

We demonstrate the effectiveness of our approach across a range of scenarios involving complex labels from diverse domains, including vision learning, e.g., semantic segmentation (Long et al., 2015; Chen et al., 2017), graph learning, e.g., constrained N-body simulation (Satorras et al., 2021), and natural language processing, e.g., supervised fine-tuning via next-token prediction in large language models (Brown, 2020; Touvron et al., 2023). The empirical results highlight the superiority of PCL over traditional supervised learning across various mainstream network backbones.

## 2. Related Work

**Learning with Deterministic Labels.** Learning with deterministic labels, often framed within the supervised learning (SL) paradigm, focuses on training models using datasets where input-output pairs are unambiguously annotated. In SL, the models typically learn the direct mapping from the data inputs to labels by minimizing the discrepancy between predicted and ground truth labels. Beyond pure supervision, hybrid frameworks integrate deterministic labels with auxiliary learning objectives, e.g., semi-supervised methods (Zhu & Goldberg, 2009) combine scarce labeled data with abundant unlabeled samples, and weakly supervised learning addresses scenarios with incomplete or noisy supervision (Zhou, 2018). For theoretical analysis of these systems, the Information Bottleneck (IB) principle (Tishby et al., 2000; Tishby & Zaslavsky, 2015) indicates that models balance the trade-off between compression and prediction accuracy. In this paper, we mainly focus on classic supervised learning scenarios to demonstrate and analyze the effectiveness of the proposed PCL learning paradigm.

Besides, different label utilization techniques have been proposed to enhance model generalization and robustness. Examples include label smoothing that replaces hard labels with soft distributions to prevent overconfidence and improve calibration (Szegedy et al., 2016; Müller et al., 2019), mixup that generates virtual training examples by interpolating both inputs and labels, augmenting data and smoothing decision boundaries (Zhang et al., 2017), curriculum learning which organizes training samples from easy to hard, effectively exploiting label difficulty to stabilize and speed up model convergence (Bengio et al., 2009; Wang et al., 2021), and focal loss that reweights the contribution of hard-to-classify examples by modulating label-related loss terms to address class imbalance (Lin et al., 2017). However, these methods still rely on using labels solely to align neural predictions, whereas we attempt to incorporate label information into the model input as a reference for learning.

**Diffusion Models and Consistency Models.** Generative models serve widespread applications for their powerful distribution learning capacity (Cheng et al., 2022; Li et al., 2022; 2023c;a; Chen et al., 2024; Guo et al., 2024). Diffusion models are characterized by a forward process of noise injection and a reverse process of learnable denoising, where neural networks iteratively predict data distributions conditioned on increasingly noisy inputs. Diffusion models have been verified in continuous space based on the Gaussian noise (Sohl-Dickstein et al., 2015; Song & Ermon, 2019; Ho et al., 2020; Song et al., 2020) and extended to discrete data with noise distributions modeled as binomial or categorical variables (Sohl-Dickstein et al., 2015; Austin et al., 2021; Hoogeboom et al., 2021). Building on the advancements of diffusion models, instead of iteratively refining noisy samples through a reverse diffusion process, consistency models (Song et al., 2023; Song & Dhariwal, 2023) leverage a self-consistency mechanism across different time steps, directly learning the mappings from noise to data in a single step. This approach has shown promise in reducing computational overhead while maintaining high sample quality. Recently, diffusion models have shown promise in more challenging data generation tasks like solving NP combinatorial problems (Sun & Yang, 2023; Li et al., 2023b; 2024; 2025; Zheng et al., 2024).

## 3. Preliminary and Background

Supervised learning aims to train the model to extract compressed features or representations of input data $\mathbf{x} \in X$ while retaining the most relevant information about the target label $\mathbf{y} \in Y$ (Tishby et al., 2000; Tishby & Zaslavsky, 2015). This is based on the assumption that the data provides sufficient information about the labels, which means the data is abundant. From an information-theoretic perspective, the mutual information $I(X; Y)$ quantifies

how much information $X$ provides about $Y$. Typically, $X$ is a high dimensional variable whereas $Y$ has a significantly lower dimensionality, which generally means that most of the entropy of $X$ is not very informative about $Y$ and the relevant features are difficult to extract (Tishby & Zaslavsky, 2015). In deep learning (LeCun et al., 2015), deep neural networks create a compressed representation $X_E$ of $X$ through an encoder, which discards irrelevant information while preserving as much of the mutual information $I(X_E; Y)$ as possible. The compression is optimized by minimizing $I(X; X_E)$ while maximizing $I(X_E; Y)$.

This formulation typically assumes that $Y$ is a low-dimensional vector (e.g., class labels) where the information content is relatively limited. However, many real-world tasks, especially in structured prediction (e.g., image segmentation, sequence generation), involve predicting high-dimensional outputs. In these tasks, the mutual information $I(X_E; Y)$ can be difficult to maximize because the high-dimensional labels themselves contain redundancies. Moreover, the space of possible outputs $Y$ could involve complex correlations that are hard to capture directly. These learning tasks with complex labels can be characterized by a label space that exhibits high complexity due to at least one of the following characteristics: (i) high dimensionality, (ii) intricate internal structure, or (iii) the presence of significant dependency patterns among labels.

In contrast to traditional tasks with simple scalar or categorical labels, complex labels encode rich, multi-dimensional, or structured information. Consequently, these tasks require models to capture sophisticated relationships and dependencies within the label space, transcending straightforward mappings from input features. In such cases, the mutual information $I(X_E; Y)$ becomes harder to maximize due to the exponential growth of label entropy $H(Y)$ with dimensionality, which further complicates the balancing between minimizing $I(X; X_E)$ and maximizing $I(X_E; Y)$. One solution to this challenge is to learn an effective latent representation $Y_E$ of the target $Y$. This concept aligns with existing approaches (Rombach et al., 2022; Hottung et al., 2021) that handle high-dimensional outputs in latent spaces. However, for prediction purposes, these methods rely on the invertibility of the mapping from $Y$ to its latent representation $Y_E$, and necessitate learning additional networks to manage latent representations. In the following section, we propose to enhance the model's ability to capture $I(X_E, Y)$ directly by leveraging the mechanism of the learning paradigm itself.

## 4. Predictive Consistency Learning

### 4.1. Intuition and Overview

To more effectively capture $I(X_E, Y)$, where $Y$ contains substantial information, directly maximizing the mutual in-

formation between $X_E$ and $Y$ can be challenging. Rather than attempting to learn all of $Y$'s information at once, we propose a structured learning process that progressively captures this information. To break down the label information into a more gradual learning process, we introduce an additional noisy label $Y_t$ to regulate the amount of label information learned at each iteration. Using $Y_t$, the original mutual information $I(X_E; Y)$ can be decomposed as:

$$I(X_E; Y) = I(X_E; Y|Y_t) + I(X_E; Y_t) - I(X_E; Y_t|Y) \tag{1}$$

Since $Y_t$ is derived from $Y$, it does not provide any more information about $X_E$ if $Y$ is given, thus the redundancy term $I(X_E; Y_t|Y) = 0$, simplifying to:

$$I(X_E; Y) = I(X_E; Y|Y_t) + I(X_E; Y_t) \tag{2}$$

The decomposition reveals two key components, where the first term, $I(X_E; Y|Y_t)$, captures the incremental information about $Y$ that can be learned given $Y_t$. This term serves as a lower bound of $I(X_E; Y)$, and the gap between them can be controlled by the information content of $Y_t$. By optimizing $X_E$ through maximizing $\mathbb{E}_t[I(X_E; Y|Y_t)]$, the model learns to progressively capture the full information content of $Y$. Specifically, when $t \to T$, $Y_t$ provides little information, and the model is forced to fully capture $I(X_E; Y)$. When $t \to 0$, $Y_t$ approximates $Y$, allowing the model to focus on refining details of the label. During training, by sampling a batch of random $t$ values, the model simultaneously learns to capture different aspects of the label. Initially, $X_E$ is expected to easily capture partial details of the label, and through iterative training, it gradually accumulates the full information content of $Y$.

For implementation, the model is exposed to noisy versions of $Y$. The inputs to the model include both $X$ and $Y_t$, where $Y_t$ serve as the condition. While introducing $Y_t$ as an auxiliary input facilitates learning, the ultimate goal is for the model to make predictions relying on $Y_t$ as little as possible. Formally, we aim to minimize the noise-conditional dependency $I_\theta(Y; Y_t|X)$, which measures the extent to which the model's predictions depend on the noisy label $Y_t$. Ideally, this term should be zero, indicating that the model's predictions are independent of $Y_t$ given $X$ and the model $\theta$. Mathematically, it can be measured as:

$$I_\theta(Y; Y_t|X) = \mathbb{E}_{X,Y_t} [D_{\mathrm{KL}} (p_\theta(Y|X, Y_t)\|p_\theta(Y|X))] \tag{3}$$

To achieve this, we introduce the prediction consistency term, which enforces $p_\theta(Y|X, Y_t, t) = p_\theta(Y|X, Y_{t'}, t')$ for all $t, t'$. This regularization ensures that the model's predictions are consistent across different noise levels, reducing the dependency on $Y_t$ and encouraging $X_E$ to encode all necessary information for accurate predictions.

The left part of the section outlines the detailed design of the proposed predictive consistency learning framework.

We start by introducing the noising processes for different labels and then thoroughly explain the training and inference scheme in the consistency learning paradigm.

## 4.2. Noise Scheduling for Labels

This section elucidates the noising processes designed to gradually incorporate noise across various label spaces.

**Diffusion on Categorical Labels.** For multi-dim categorical labels in $\{1, \cdots, K\}^N$ where $K$ denotes the category number and $N$ denotes the dimension, we follow discrete diffusion models (Sohl-Dickstein et al., 2015; Austin et al., 2021; Hoogeboom et al., 2021) to model the noising process as introducing multinomial noise to the label at each timestep $t$. We represent the label as $\mathbf{y} \in \{0, 1\}^{N \times K}$, which is a concatenation of $N$ one-hot vectors. The noise can be understood as transitioning between different categories for each of the $N$ dimensions. Starting from the initial point $\mathbf{y}_0 = \mathbf{y}$, the noising process is defined as:

$$q(\mathbf{y}_t|\mathbf{y}_{t-1}) = \mathrm{Cat}(\mathbf{y}_t; \mathbf{p} = \mathbf{y}_{t-1}\mathbf{Q}_t), \tag{4}$$

where $\mathrm{Cat}(\mathbf{y}; \mathbf{p})$ is categorical distributions over $N$ one-hot vectors with probabilities given by $\mathbf{p}$, and $\mathbf{Q}_t = (1 - \beta_t)\mathbf{I} + \beta_t/K \mathbf{1}\mathbf{1}^\top \in \mathbb{R}^{K \times K}$ is the transition matrix, which determines the corruption introduced at timestep $t$, where $\beta_t$ is the corruption rate at timestep $t$. This ensures that with probability $\beta_t$, the corresponding label category can transition to any other category. Over time, as $t$ approaches the final timestep $T$, the labels converge towards a uniform distribution over the $K$ categories. The cumulative effect of the diffusion process after $t$ steps is:

$$q(\mathbf{y}_t|\mathbf{y}_0) = \mathrm{Cat}(\mathbf{y}_t; \mathbf{p} = \mathbf{y}_0\bar{\mathbf{Q}}_t), \tag{5}$$

where $\bar{\mathbf{Q}}_t = \mathbf{Q}_1\mathbf{Q}_2 \ldots \mathbf{Q}_t$ represents the accumulated transition matrix from $\mathbf{y}_0$ to $\mathbf{y}_t$.

**Diffusion on Continuous Labels.** For multi-dim continuous labels in $\mathbb{R}^N$, where $N$ denotes the dimensionality, we follow Gaussian diffusion models (Sohl-Dickstein et al., 2015; Ho et al., 2020; Nichol & Dhariwal, 2021) to model the diffusion process as introducing Gaussian noise to the label at each timestep. At each timestep $t$, Gaussian noise is applied to corrupt the label, progressively pushing it toward a noisy distribution. The noising process is defined as:

$$q(\mathbf{y}_t|\mathbf{y}_{t-1}) = \mathcal{N}(\mathbf{y}_t; \sqrt{1 - \beta_t}\mathbf{y}_{t-1}, \beta_t\mathbf{I}), \tag{6}$$

where $\mathcal{N}(\mathbf{y}; \mu, \Sigma)$ is a Gaussian distribution with mean $\mu$ and covariance $\Sigma$, and $\beta_t$ controls the variance of the added noise at timestep $t$. The factor $\sqrt{1 - \beta_t}$ ensures that the label retains some of its original value, while the noise is introduced with variance $\beta_t$. Over time, as $t$ approaches the final timestep $T$, the labels converge towards a Gaussian centered at zero. The cumulative effect of this diffusion process

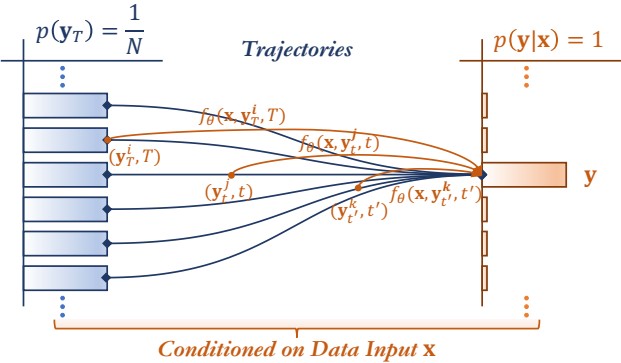

*Figure 2.* Prediction consistency enforces all trajectories conditioned on **x** consistently map to the same initial point, i.e. label **y**.

after $t$ steps is described by the marginal distribution:

$$q(\mathbf{y}_t|\mathbf{y}_0) = \mathcal{N}(\mathbf{y}_t; \sqrt{\bar{\alpha}_t}\mathbf{y}_0, (1 - \bar{\alpha}_t)\mathbf{I}), \qquad (7)$$

where $\bar{\alpha}_t = \prod_{i=1}^{t}(1 - \beta_i)$ is the accumulated noise scale from the original label $\mathbf{y}_0$ to the noisy label $\mathbf{y}_t$. As $t$ increases, $\bar{\alpha}_t$ decreases, leading to increased label corruption.

**Diffusion on Embeddings.** In scenarios where the label is too complex to be directly expressed as a categorical or continuous value, or when the number of categories is excessively large, we introduce Gaussian noise directly to the latent embeddings of the labels in a manner consistent with the noising process used for continuous labels.

### 4.3. Predictive Consistency Training Scheme

We draw inspiration from the consistency mapping concept in consistency models (Song et al., 2023; Song & Dhariwal, 2024) and define the predictive consistency mapping for prediction scenarios as $f_\theta : (\mathbf{x}, \mathbf{y}_t, t) \mapsto \mathbf{y}$. Corresponding to the discussions in Sec. 4.1, we expect the training scheme to satisfy two properties: 1) $f_\theta$ can nearly recover the label **y** given the inputs $\mathbf{x}, \mathbf{y}_t, t$, where $t$ informs the model of the noise level; 2) the prediction consistency that $f_\theta(\mathbf{x}, \mathbf{y}_t, t) = f_\theta(\mathbf{x}, \mathbf{y}_{t'}, t')$ holds for all $t, t'$. The key distinction between the proposed predictive consistency mapping and the generative consistency mapping is that in the former, the diffusion trajectories are conditioned on the data input **x** with a reference target label **y**. PCL aims to recover the exact **y** given **x**, where the target distribution converges to an exact target point, and the model trades the output diversity to better capture **y**. This requires the prediction to focus on the condition **x**, with predictive consistency enforced across all possible label noising trajectories instead of satisfying self-consistency (Song et al., 2023) within merely each generative trajectory. The complete predictive consistency condition can be expressed as follows: conditioned on data input **x**, all points along any trajectory map to its

label, i.e., $f_\theta(\mathbf{x}, \mathbf{y}_t^i, t) = f_\theta(\mathbf{x}, \mathbf{y}_{t'}^j, t') = \mathbf{y}$ for distinct trajectories $i$ and $j$ (from independent noising processes) at distinct steps $t$ and $t'$, as shown in Fig. 2.

To achieve the consistency to learn $f : \mathbf{x} \mapsto \mathbf{y}$, given that the target **y** is certain and explicit, we do not have to merely rely on optimizing the expectation of the variation of the consistency mappings over two noise points $\mathbf{y}_t$ and $\mathbf{y}_{t'}$ to propagate the label information across different noise levels as generative consistency models do (Song et al., 2023). Instead, we additionally introduce **y** to optimize the triadic distance to achieve prediction consistency:

$$\mathcal{L}_{\text{PCL}}(\theta) = \mathbb{E}\big[\lambda_1 d\big(f_\theta(\mathbf{x}, \mathbf{y}_t, t), \mathbf{y}\big) + \lambda_1 d\big(f_\theta(\mathbf{x}, \mathbf{y}_{t'}, t'), \mathbf{y}\big) \\ + \lambda_2 d\big(f_\theta(\mathbf{x}, \mathbf{y}_t, t), f_\theta(\mathbf{x}, \mathbf{y}_{t'}, t')\big)\big]. \tag{8}$$

Here $d(\cdot, \cdot)$ is a distance metric function and $\lambda_1, \lambda_2$ are loss weights. Specifically, to align with the traditional supervised training paradigm, we retain the original task-defined loss function for the distance metric $d$, such as cross-entropy for classification tasks and mean squared error for regression tasks. This is because the design of the loss function is orthogonal to our learning framework, allowing them to complement each other. The main modification in our approach lies in that the model predicts **y** based on both **x** and the noise-perturbed versions of **y**, while ensuring predictive consistency across different noise levels. In practice, rather than randomly sampling two timesteps $t_1$ and $t_2$, we explicitly control the noise gap between them by setting $t_2 = \alpha t_1$. This ensures that the predictive consistency remains a strong and informative constraint to learn. We then independently sample from the noise distribution to obtain $\mathbf{y}_{t_1}^i$ and $\mathbf{y}_{t_2}^j$ to ensure that the two noisy samples are independent with respect to both the time steps and the diffusion trajectories. Then Eq. 8 can be effectively optimized to learn the consistency predictive mapping, and the whole training process is presented in Alg. 1 and Fig. 1.

### 4.4. Multistep Inference with Consistency Mappings

With a well-trained $f_\theta(\cdot, \cdot, \cdot)$, we obtain predictions for a given **x** by sampling $\mathbf{y}_T$ from the uniform distribution and then evaluate $\mathbf{y}_0 = f_\theta(\mathbf{x}, \mathbf{y}_T, T)$. This standard single-step inference requires only one forward pass through the model, akin to conventional direct predictions. During training, the accuracy tends to be higher when $t$ is small, as the label hints contain a richer amount of information. Our objective is to progressively transfer this high accuracy to larger values of $t$ through training, thereby enhancing overall model performance. In the ideal case that the consistency loss converges to zero, optimal results can be achieved in a single step, yet in practice, gradually decreasing $t$ from $T$ to $0$ can lead to accuracy improvements. To achieve such enhancements, a multistep inference strategy can be adopted, which iteratively alternates between denoising and reintroducing noise.

---

**Algorithm 1** Predictive Consistency Training

---

1: **Input:** Dataset $\mathcal{D}$, model $f_\theta$, noise function $q(\cdot)$, learning rate $\eta$, loss weights $\lambda_1, \lambda_2$, noise gap $\alpha$
2: **repeat**
3:     Sample $(\mathbf{x}, \mathbf{y}) \sim \mathcal{D}$, and $t_1 \sim \mathrm{U}[1, T]$, $t_2 = \alpha t_1$
4:     Sample $\mathbf{y}_{t_1} \sim q(\mathbf{y}_{t_1}|\mathbf{y})$, $\mathbf{y}_{t_2} \sim q(\mathbf{y}_{t_2}|\mathbf{y})$
5:     $\hat{\mathbf{y}}_0^{t_1} \leftarrow f_\theta(\mathbf{x}, \mathbf{y}_{t_1}, t_1)$
6:     $\hat{\mathbf{y}}_0^{t_2} \leftarrow f_\theta(\mathbf{x}, \mathbf{y}_{t_2}, t_2)$
7:     $\mathcal{L} \leftarrow \lambda_1 d(\hat{\mathbf{y}}_0^{t_1}, \mathbf{y}) + \lambda_1 d(\hat{\mathbf{y}}_0^{t_2}, \mathbf{y}) + \lambda_2 d(\hat{\mathbf{y}}_0^{t_1}, \hat{\mathbf{y}}_0^{t_2})$
8:     $\theta \leftarrow \theta - \eta \nabla_\theta \mathcal{L}$
9: **until** convergence

---

**Algorithm 2** Multistep Prediction

---

**Input:** model $f_\theta$, data input $\mathbf{x}$, noise function $q(\cdot)$, time steps $\tau_1 > \tau_2 > \cdots > \tau_{N_\tau - 1}$, preset inference step $N_i$

Sample random noise $\mathbf{y}_T$
$\hat{\mathbf{y}}_0 \leftarrow f_\theta(\mathbf{x}, \mathbf{y}_T, T)$
**for** $n = 1$ to $N_i - 1$ **do**
    Sample $\mathbf{y}_{\tau_n} \sim q(\mathbf{y}_{\tau_n}|\hat{\mathbf{y}}_0)$
    $\hat{\mathbf{y}}_0 \leftarrow f_\theta(\mathbf{x}, \mathbf{y}_{\tau_n}, \tau_n)$
**end for**
**Output:** Prediction $\hat{\mathbf{y}}_0$

---

This approach effectively trades off runtime for enhanced prediction quality, allowing the model to refine its outputs over multiple inference steps and leverage increasingly rich information embedded in earlier predictions.

Given a sequence of time points $\tau_1 > \tau_2 > \cdots > \tau_{N_\tau - 1}$, at each step $\tau_n$, the current prediction $\mathbf{y}_{\tau_{n-1}}$ is perturbed by a noise function to a state $\mathbf{y}_{\tau_n}$. The noise level decreases with each step, meaning $\tau_n < \tau_{n-1}$. The model then denoises the corrupted label by applying $f_\theta(\mathbf{x}, \mathbf{y}_{\tau_n}, \tau_n)$, producing a refined prediction. This process is repeated over successive steps, where each newly refined label incorporates progressively more accurate information from the previous step. This enables the model to gradually recover the whole information of $\mathbf{y}$ by taking the perhaps approximated prediction as the label hints and leveraging the incrementally informative hints for the final prediction.

In practice, we observe that as the number of inference steps increases, the model's prediction error initially continues decreasing, but may then rise for a period, as shown in Fig. 3. This phenomenon may result from the fact that during training, the model is only exposed to the noise-perturbed versions of ground truth labels as inputs. However, during inference, the intermediate predictions of the model may contain errors, which can accumulate over multiple steps of inference. Consequently, there exists a tradeoff between gradually capturing finer prediction details and accumulating prediction errors. To address this, we set the

total number of time steps to 10 and use the validation set to identify the step that best balances this tradeoff and achieves the highest prediction accuracy. During testing, we perform early stopping at this identified inference step. The specific multistep prediction procedure is in Alg. 2.

## 5. Experiments

Since PCL is proposed as a novel training paradigm, the major baseline for comparison is conventional supervised learning. The comparisons are conducted across various classic and representative model backbones for different modalities to showcase the general applicability of PCL. We test the proposed PCL framework on tasks involving complex labels from diverse domains, including semantic segmentation (high-dimensional categorical outputs in vision learning), N-body simulation (high-dimensional continuous outputs in graph learning), and next-token prediction (high-dimensional sequential outputs in language modeling).

Experiments for constrained n-body simulation are conducted on a single GPU of NVIDIA RTX 4090. For semantic segmentation, a single NVIDIA H100 GPU was employed, and experiments for next-token prediction are performed on 8 GPUs of NVIDIA H800.

### 5.1. Constrained N-body Simulation

The constrained N-body simulation (Huang et al., 2022) task involves predicting the future positions of a set of interacting particles over time based on initial conditions such as their positions, velocities, and their interactions from constraints like sticks and hinges indicating connection relations, as well as the inherent physical forces. The evolution of particle positions and velocities follows physical laws. We incorporate tasks with varying complexities by controlling the number of isolated particles $p$, the number of sticks $s$ and the number of hinges $h$. The configuration is abbreviated as $(p, s, h)$ and we follow (Huang et al., 2022) to incorporate five tasks corresponding to configurations of $(1, 2, 0)$, $(2, 0, 1)$, $(3, 2, 1)$, $(0, 10, 0)$, and $(5, 3, 3)$.

**Dataset.** We collect 5000 trajectories for training, 2000 for validation, and 2000 for testing for each configuration. Each trajectory spans 1000 timesteps. For each trajectory, the initial conditions include the particle positions $p(0) \in \mathbb{R}^{p \times 3}$, the initial velocities $v(0) \in \mathbb{R}^{p \times 3}$, and the respective charges $c \in \{-1, 1\}^p$. There exist stick and hinge connections among the particles. The task is to predict the positions of the five particles after 1000 timesteps.

**Metrics.** We evaluate the prediction error by Mean Square Error (MSE), i.e., the average of the squares of the errors between the predicted values and the true values.

**Model Design.** We consider the state-of-the-art graph mod-

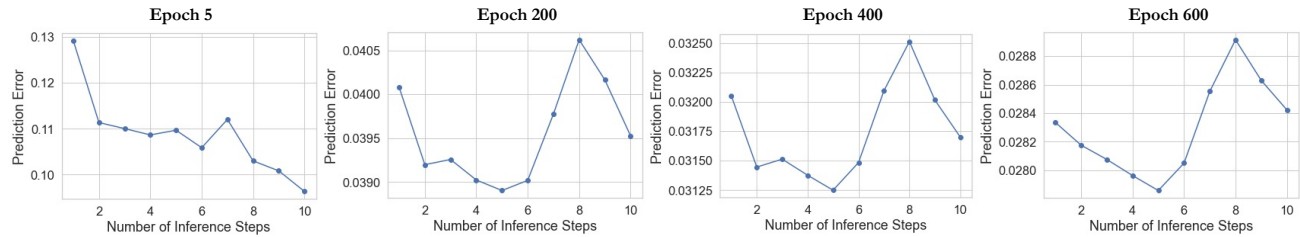

*Figure 3.* The influence variation of inference steps across the training process.

*Table 1.* Prediction error ($\times 10^{-2}$) of SL and PCL on top of graph models on various types of N-body simulation systems. The header of each column "$p, s, h$" denotes the scenario with $p$ isolated particles, $s$ sticks and $h$ hinges.

| Backbone Model | Training | $1, 2, 0$ | $2, 0, 1$ | $3, 2, 1$ | $0, 10, 0$ | $5, 3, 3$ |
|---|---|---|---|---|---|---|
| GCN | SL | 2.865±0.021 | 2.534±0.061 | 3.479±0.110 | 4.705±0.046 | 4.303±0.002 |
| | PCL | **2.436±0.040** | **2.268±0.012** | **2.795±0.061** | **3.162±0.251** | **3.228±0.221** |
| GAT | SL | 2.921±0.198 | 2.707±0.024 | 3.351±0.111 | 3.478±0.342 | 3.407±0.180 |
| | PCL | **2.771±0.208** | **2.581±0.026** | **2.802±0.216** | **2.481±0.059** | **2.534±0.011** |
| GGNN | SL | 3.013±0.022 | 2.716±0.068 | 3.293±0.023 | 4.426±0.044 | 4.148±0.035 |
| | PCL | **2.614±0.031** | **2.297±0.033** | **2.974±0.011** | **3.191±0.290** | **3.457±0.213** |

*Table 2.* Ablation study on loss construction on top of different backbones reflected by prediction error ($\times 10^{-2}$).

| | GCN | GAT | GGNN |
|---|---|---|---|
| Traditional SL | 3.479±0.110 | 3.351±0.111 | 3.293±0.023 |
| w/o $\lambda_1$-term, w/ $\lambda_2$-term | 6.784±0.862 | 4.234±0.170 | 3.731±0.915 |
| w/ $\lambda_1$-term, w/o $\lambda_2$-term | 2.881±0.110 | 3.001±0.074 | 3.045±0.090 |
| w/ $\lambda_1$-term, w/ $\lambda_2$-term | **2.795±0.061** | **2.802±0.216** | **2.974±0.011** |

eling solution for this task, and the model is optimized by minimizing the averaged MSE between the predicted positions and the ground truth positions. We input the concatenation of the initial positions and the velocities as the node features. The edge feature is provided by a concatenation of the product of charges $c_i c_j$ and an edge type indicator $I_{ij}$, where $I_{ij}$ is valued as 0 if node $i$ and $j$ are disconnected, 1 if connected by a stick, and 2 if connected by a hinge. We take the model outputs as the estimated positions. To introduce PCL, we adopt two linear layers to encode the input attributes and the noised label, respectively, and then concatenate them to form the input hidden feature to the subsequent graph neural layers. We adopt 4 graph neural layers, and for each layer's output, we integrate the timestep embedding extracted by the sinusoidal position embedding and a linear layer through addition. In this task, $\mathbf{y} \in \mathbb{R}^{p \times 3}$ and we adopt the Gaussian noising process to produce noised labels as shown in Eq. 7.

**Results.** For graph modality, we compare PCL and the conventional supervised learning on top of classic graph neural networks, including Graph Convolutional Networks (GCN) (Kipf & Welling, 2016), Graph Attention Network

(GAT) (Veličković et al., 2017), and Gated Graph Neural Networks (GGNN) (Li et al., 2015). Table 1 shows the superiority of PCL on quantitative results with significantly lower estimation error across five tasks with different complexity. As can be discovered, the performance gains are more pronounced on more complex datasets.

Table 2 provides ablation on the effects of the $\lambda_1$-term and $\lambda_2$-term in Eq. 8. $\lambda_2$-term corresponds to the design from the generative consistency models that enforce the self-consistency loss, which is less suitable for this scenario. Compared to merely adopting the $\lambda_1$-term, $\lambda_2$-term can serve as a regularization term to more directly enforce predictive consistency, resulting in certain performance gains.

Fig. 3 illustrates the testing performance across different inference steps in a 10-step prediction process. In the early stages, increasing the number of inference steps clearly reduces prediction error. However, as training progresses, the trend becomes more variable, first decreasing, then increasing, and finally decreasing again. This behavior may stem from the trade-off between capturing finer details and the accumulation of errors as discussed in Sec. 4.4.

### 5.2. Semantic Segmentation

Semantic segmentation is a classic dense vision task that involves classifying each pixel of an image into a predefined category (Long et al., 2015; Zhao et al., 2017). Unlike classification tasks that categorize entire images, semantic segmentation analyzes the finer granularity of images to identify the boundaries and relationships between objects.

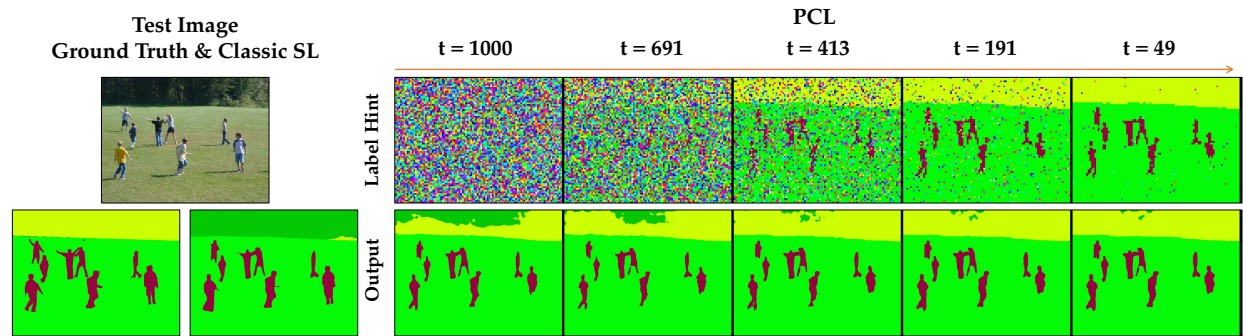

*Figure 4.* Predictions across varying timesteps based on the last step's predictions in the multistep inference procedure. In each step, the model receives the input and label hint and predicts the output.

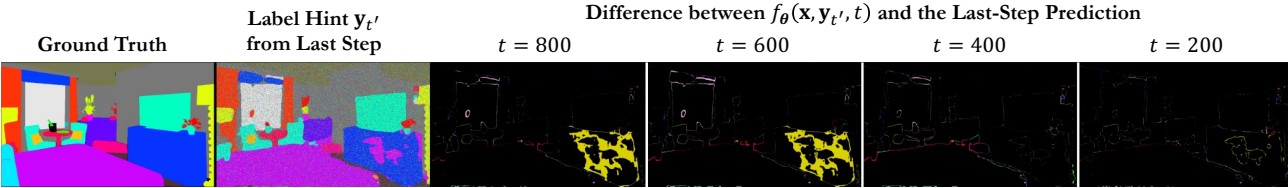

*Figure 5.* The visualization of the prediction improvements with different $t$ controlling the prediction granularity.

It plays a critical role in various applications, such as autonomous driving (Badrinarayanan et al., 2017), medical imaging (Ronneberger et al., 2015), and scene understanding (Long et al., 2015).

**Dataset.** We utilize the ADE20K dataset (Zhou et al., 2019), which is a commonly used large-scale scene parsing dataset that contains over 20K images with pixel-level annotations. The dataset is annotated with 150 different object classes, and we make the unannotated pixels as -1, which are ignored during training and testing. Following Zhou et al. (2017; 2019), we resize the original images during training while maintaining a constant aspect ratio, randomly scaling the shorter side to one of the sizes: 300, 375, 450, 525, or 600.

**Metrics.** Following Zhou et al. (2017; 2019), we adopt three evaluation metrics to measure model performance: 1) Pixel Accuracy: the proportion of correctly classified pixels. 2) Mean IoU (mIoU): the intersection-over-union between the predicted and ground-truth pixels, averaged over all the classes. 3) Score: the average value of Pixel Accuracy and Mean IoU. During the testing phase, we use Multi-Scale Test: evaluate at multiple sizes and then take the average.

**Model Design.** We adopt an encoder-decoder framework. The encoder compresses the input by extracting high-level features using a CNN backbone, reducing the spatial resolution while capturing important semantic information. The decoder then progressively upsamples the compressed features to recover the original resolution. To introduce PCL, we first downsample the noised labels processed by the embedding layer using the timestep embeddings extracted through sinusoidal position embedding. These down-

*Table 3.* Results on Semantic Segmentation.

| Backbone Model | Training | Pixel Acc.↑ | mIoU↑ | Score↑ |
|---|---|---|---|---|
| MobileNetV2dilated | SL | 75.53 | 33.13 | 54.33 |
| | PCL | **76.42** | **35.27** | **55.85** |
| ResNet50dilated+PPM | SL | 78.98 | 41.49 | 60.24 |
| | PCL | **82.33** | **48.56** | **65.45** |

sampled labels are then fused with the image features extracted by the encoder. Finally, the fused image features are fed into the decoder for further processing. In this task, $\mathbf{y} \in \{-1, ..., 149\}^{H \times W}$ where each entry will be converted into a one-hot vector of length 151, indicating the classification of pixels. We adopt the categorical noising process in Eq. 5 using transition matrices of $\mathbf{Q}_t \in [0, 1]^{151 \times 151}$.

**Results.** For the encoder, we choose ResNet50dilated (He et al., 2016) and MobileNetV2dilated (Sandler et al., 2018). For the decoder, we sequentially selected PPM (Pyramid Pooling Module) (Zhao et al., 2017) with DeepSup (deep supervision trick) and C1 (one convolution module) with DeepSup. Table 3 shows PCL's superiority over SL. Fig. 4 visually demonstrates how increasing inference steps further improves predictions, particularly for large background areas. Fig. 5 shows the prediction improvements on top of the last-step prediction with varying $t$. Compared to the previous step's prediction $\hat{\mathbf{y}}_0$ with its noised version $\mathbf{y}_{t'}$ as the label hint, for the prediction $f_\theta(\mathbf{x}, \mathbf{y}_{t'}, t)$, setting a larger $t$ tends to encourage the model to improve broader structural relationships, while setting a smaller $t$ encourages the model to focus on finer details, such as object boundaries.

## 5.3. Supervised Next-token Prediction Fine-tuning

The next-token prediction task is a cornerstone in natural language processing, forming the foundation for transformer-based large language models (LLMs) such as GPT (Radford, 2018), LLaMA (Touvron et al., 2023). The objective of the task is to predict the next token in a sequence, given the preceding context. The task's outputs retain similar dimensionality and sequence complexity as the inputs. Models trained on next-token prediction tasks have proven to be highly effective at capturing both short-term and long-range dependencies in language, enabling them to generate coherent, contextually appropriate text. This section investigates the effectiveness of PCL in the full fine-tuning task on the pre-trained LLaMa-2-7B (Touvron et al., 2023) models.

**Dataset and Metrics.** The Alpaca (Taori et al., 2023) dataset is based on the self-instruct method (Wang et al., 2022), utilizing the OpenAI text-davinci-003 engine to generate a collection of instructions and demonstrations. These instruction data can be employed for fine-tuning language models. By filtering out low-quality data, such as hallucinations, incorrect answers, and unclear instructions, we obtain the Alpaca-cleaned dataset, which serves as the sole training data for all models discussed in this paper.

To evaluate the performance of models, we employ INSTRUCTEVAL (Chia et al., 2023), a comprehensive evaluation suite designed specifically for instruction-tuned models. INSTRUCTEVAL aims to assess models across dimensions such as problem-solving ability, writing proficiency, and alignment with human values. Following INSTRUCTEVAL, the evaluation of large language model performance in this paper includes benchmarks:

- MMLU (Hendrycks et al., 2020) assesses models' world knowledge and reasoning abilities across 57 academic disciplines. Questions range in difficulty, presented in a multiple-choice format. The evaluation primarily uses few-shot settings to test the generalization capabilities. A 5-shot direct prompting is utilized for evaluation.

- BBH (Srivastava et al., 2022) is a subset of 23 challenging tasks from the BIG-Bench benchmark. It evaluates the ability to handle challenging reasoning and problem-solving tasks that go beyond simple language understanding. We apply 0-shot direct prompting to measure the model's capability in dealing with unseen questions without additional contextual examples.

- CRASS (Frohberg & Binder, 2022) evaluates models' ability to handle complex relational reasoning tasks, specifically in the context of causal structures and relationships. It includes a variety of problems that test how well the model understands and predicts causal

*Table 4.* Evaluation on LLM fine-tuning. Relative performance improvements compared to the raw model are marked in brackets.

| Backbone | Training | MMLU | CRASS | BBH |
|----------|----------|------|-------|-----|
| Raw Model | – | 41.90 | 37.59 | 32.93 |
| Full FT | SL | 46.22 | 58.29 | 33.38 |
| | PCL | **47.10** | **59.48** | **34.75** |

relationships between different entities or events. We use 3-shot direct prompting to assess model reasoning.

**Model Design.** Our modified LLaMA2-7B retains the original embedding and decoder layers but introduces a novel mechanism to predict the next token. Instead of the standard approach where hidden states directly generate token probabilities, we inject Gaussian noise into token embeddings in training to enhance robustness. This noise is combined with the tokens hidden state to form an augmented vector, which is passed via an MLP and classification head for prediction. Additionally, we enforce consistency by minimizing the mean squared error between logits at randomly selected time steps. In the generation phase, noise is iteratively reduced across steps to generate tokens sequentially. More detailed descriptions can be found in Appendix B.3.1.

**Results.** We compare methods for fine-tuning LLMs on the next-token prediction task. The baselines include the pre-trained LLaMA2-7B model and the LLaMA2-7B model fine-tuned using traditional full-parameter supervised learning (Taori et al., 2023). In contrast, our method also tunes the full parameters of the model but operates under the predictive consistency learning paradigm. Table 4 shows the superiority of PCL on quantitative results across MMLU, GRASS and BBH.

## 6. Conclusion and Future Work

This paper has proposed a novel predictive consistency learning framework beyond previous methods with direct prediction, aiming to explore the full potential of label information for supervision during the learning process. It has shown superiority in experiments on various tasks across vision, text, and graph modalities with complex labels. It resembles, to a certain degree, the residual learning scheme by treating the noisy label as the input, which serves as a counterpart to the raw signal of the input data. Future work will explore the potential of applying label diffusion to other fundamental deep learning paradigms, such as semi-supervised and weakly supervised learning. It will also explore more effective label granularity, e.g., treating entire sentences as label hints rather than individual tokens, and design more natural label noising mechanisms to enhance learning.

## Impact Statement

This work introduces a novel training paradigm that fundamentally rethinks the traditional direct mapping approach in supervised learning and beyond. Specifically designed to address the growing challenge of capturing complex, high-dimensional labels in modern prediction tasks, the method decomposes full label information into a progressive learning procedure and leverages noise-perturbed labels as additional references. As a generalizable framework, PCL has the potential to fundamentally improve a wide range of applications in a way that is orthogonal to network architectures and loss function designs. Beyond its practical impact, this work also serves as an exploration to answer a fundamental question: *When labels contain rich information, can they be leveraged to facilitate learning rather than simply being treated as prediction targets?* By demonstrating the effectiveness of incorporating label information into model input for reference, this study opens new avenues for rethinking how labels are utilized in machine learning.

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

# Appendix

# A. More Discussions

## A.1. Computational Overhead in Training and Inference

For training, since the loss calculation requires two inference predictions with different noise levels during training, it requires twice the training cost of the traditional supervised learning paradigm. Moreover, it typically takes less than twice the iterations for the model to converge.

For inference, PCL can produce much superior predictions with merely a single forward pass as SL does. Meanwhile, PCL can achieve better performance with more forward passes. As shown in Fig. 3 using a GCN backbone on the 3,2,1 dataset, even single-step PCL inference achieves superior performance (prediction error 0.02835) compared to SL (0.03478), while additional inference steps (e.g., 5-step) can further refine results (0.02795).

## A.2. Comparison to Curriculum Learning Strategies that Gradually Increase Label Complexity

The key differences between predictive consistency learning (PCL) and curriculum learning (CL) can be summarized as follows:

- Learning Target Consistency. CL predefines staged learning targets (e.g., coarse-to-fine labels), where each stage learns an approximation of the true label. This risks error accumulation, i.e., biased features from early stages may propagate to later stages. In contrast, the objective of PCL remains stable: always predicting the true label, with complexity dynamically adjusted by the input noise level.

- Progressive vs. Simultaneous Learning. CL follows a fixed, sequential progression (e.g., easyhard labels). PCL, however, randomly samples time steps during training, enabling the model to learn various label information levels (from partial to complete) simultaneously, where the time step t acts as a controller for label granularity.

- Controllable Prediction. PCLs noise-conditioned framework allows explicit control over prediction granularity via time step t. As evidenced in Fig. 5, setting a larger t tends to encourage the model to improve broader structural relationships, while setting a smaller t encourages the model to focus on finer details. Further, PCL supports multi-step inference (Sec. 4.4), where predictions are iteratively refined (analogous to diffusion denoising), boosting final accuracy.

- Leveraging Partial Labels as Input. One of the motivations of PCL is to treat labels not just as targets but as learning facilitators. By feeding partially noised labels (e.g., "hints") during training, the model learns to exploit intermediate information (similar to how humans use reference solutions to help solve problems).

# B. Experiment Details

## B.1. Experimental Details for N-body Simulation

### B.1.1. NOISING PROCESS

At timestep $t = 0$, the label $\mathbf{y} \in \mathbb{R}^{5\times3}$ represents the original 3-dimensional coordinates of the 5-body system. We introduce Gaussian noise that gradually transforms the coordinates to points from the standard Gaussian distributions. The noising process simply follows Eq. 6 and Eq. 7.

### B.1.2. MODEL ARCHITECTURE

We follow the implementation of (Satorras et al., 2021) to generally implement 4-layer GNNs. With its learnable edge operation function $\phi_e$ and node operation function $\phi_h$, the graph convolutional layer follows:

$$\mathbf{m}_{ij} = \phi_e(\mathbf{h}_i^l, \mathbf{h}_j^l, a_{ij}) \tag{9}$$

$$\mathbf{m}_i = \sum_{j \in \mathcal{N}(i)} \mathbf{m}_{ij} \tag{10}$$

$$\mathbf{h}_i^{l+1} = \phi_h(\mathbf{h}_i^l, \mathbf{m}_i) \tag{11}$$

Where $\mathbf{h}_i^l \in \mathbb{R}^{\mathrm{nf}}$ is the nf-dimensional embedding of node $v_i$ at layer $l$. $a_{ij}$ are the edge attributes. $\mathcal{N}(i)$ represents the set of neighbors of node $v_i$. Here, $\phi_e$ and $\phi_h$ are approximated by 2-layer Multilayer Perceptrons (MLPs).

The initial position $\mathbf{p}^0$ and velocity $\mathbf{v}^0$ from the particles are passed through a linear layer to obtain the input feature. The label hint is passed through another linear layer, and the obtained feature is concatenated with the input feature and inputted into the GNN first layer $\mathbf{h}^0$. The particle's charges are inputted as edge attributes $a_{ij} = c_i c_j$. The time step $t$ is first embedded through the sinusoidal position embedding

$$\tilde{\mathbf{t}} = \mathrm{concat}\left(\sin\frac{t}{T^{\frac{0}{d}}}, \cos\frac{t}{T^{\frac{0}{d}}}, \sin\frac{t}{T^{\frac{2}{d}}}, \cos\frac{t}{T^{\frac{2}{d}}}, \ldots, \sin\frac{t}{T^{\frac{d}{d}}}, \cos\frac{t}{T^{\frac{d}{d}}}\right) \tag{12}$$

and then processed through linear layers and activation functions. Here $d$ is the embedding dimension, $T$ is a large number (usually selected as 10000), $\mathrm{concat}(\cdot)$ denotes concatenation. Then we aggregate the timestep feature with the node convolutional feature and reformulate the update for node features, i.e., Eq. 11 as:

$$\mathbf{h}_i^{l+1} = \phi_h(\mathbf{h}_i^l, \mathbf{m}_i) + \phi_t(\tilde{\mathbf{t}}) \tag{13}$$

where $\phi_h$ is a linear layer model. The output of the GNN $\mathbf{h}^L$ is passed through a two layers MLP that maps it to the estimated position.

## B.2. Experimental Details for Semantic Segmentation

Given image information $\mathbf{x}$ extracted through the encoder, hint label $\mathbf{y}$, and time step $t$, PCL first embeds the latter two and then down-sample the the embedded hint label $\tilde{\mathbf{y}}$ with the time step embeddings $\tilde{\mathbf{t}}$ to accommodate the dimensions of $\mathbf{x}$. Then PCL merges the image features and the noise labels containing time step information into a new image information $\tilde{\mathbf{x}}$.

### B.2.1. EMBEDDINGS OF HINT LABEL AND TIME STEP

The hint label for semantic segmentation is generated by adding categorical noise to the ground truth labels. Given the hint $\mathbf{y}$, it is first passed through an embedding layer that maps each class $\mathbf{y} \in \{0, 1, \ldots, C\}$ (where $C$ is the number of classes) to a higher-dimensional vector $\tilde{\mathbf{y}}$. $\tilde{\mathbf{y}}$ is then processed through linear layers.

Time step $t$ is first embedded through the sinusoidal position embedding and then processed through linear layers and activation function as below, where $d$ is the embedding dimension, $T$ is a large number (usually selected as 10000).

$$\tilde{\mathbf{t}} = \mathrm{concat}\left(\sin\frac{t}{T^{\frac{0}{d}}}, \cos\frac{t}{T^{\frac{0}{d}}}, \sin\frac{t}{T^{\frac{2}{d}}}, \cos\frac{t}{T^{\frac{2}{d}}}, \ldots, \sin\frac{t}{T^{\frac{d}{d}}}, \cos\frac{t}{T^{\frac{d}{d}}}\right) \tag{14}$$

### B.2.2. DOWN-SAMPLE

To perform down-sampling of the hint label $\tilde{\mathbf{y}}$ using the time step embedding in the provided code, we first process them through two residual blocks as below. Then the hint label is then down-sampled using a convolution with a stride of 2.

$$\tilde{\mathbf{y}} = \mathrm{Conv2D}\left(\mathrm{Conv2D}\left(\mathrm{SiLU}(\mathrm{GN}(\tilde{\mathbf{y}}))\right) + \mathrm{Linear}\left(\mathrm{SiLU}(\tilde{\mathbf{t}})\right)\right) \tag{15}$$

, where $\mathrm{GN}(\cdot)$ denotes group normalization, $\mathrm{SiLU}(\cdot)$ is the sigmoid-weighted linear unit activation function, $\mathrm{Conv2D}(\cdot)$ represents a 2D convolution operation, and $\mathrm{Linear}(\cdot)$ is a fully connected layer that transforms the time step embedding.

## B.3. Experimental Details for Next-token Prediction

### B.3.1. MODEL DESIGN

In our approach, we adopt the LLaMA2-7B model as the backbone, preserving the structure of the embedding and decoder layers. However, we modify the prediction mechanism for the next token using the hidden states. In a conventional decoder-only language model (LLM), the prediction of the next token $y$ is achieved by leveraging the preceding context, encoded in a high-dimensional hidden state $h$. This hidden state $h$ is then passed through a linear layer, typically referred to as the language modeling head (lm_head) or unembedding layer, to yield the probability distribution $P(y)$ over the next token.

In contrast, our model introduces a noise injection mechanism to perturb the token embeddings, aiming to enhance robustness and generalization.

**Training Phase.** During the training phase, when the model obtains the last hidden states for each token, instead of directly passing them through the lm_head to generate logits and compute the cross-entropy loss with the ground truth labels $\hat{y}$, we transform these labels back into its corresponding embedding $y_{emb}$. Then add Gaussian noise $\epsilon \sim \mathcal{N}(0, \sigma^2)$ to $y_{emb}$, resulting in a perturbed embedding:

$$\mathbf{y}_t^{\text{noisy}} = \bar{\alpha}_t \mathbf{y}^{\text{emb}} + \bar{\beta}_t \epsilon. \tag{16}$$

To inform the model of the noise magnitude, we also perturb the time step $t$, obtaining a corresponding time embedding $\mathbf{t}_{emb}$. The noisy token embedding $\mathbf{y}_t^{\text{noisy}}$ is combined with $\mathbf{t}_{emb}$ to form a new noisy information vector,

$$\mathbf{h}_t^{\text{noisy}} = \mathbf{y}_t^{\text{noisy}} + \mathbf{t}^{\text{emb}}. \tag{17}$$

This noisy information is concatenated with the hidden state $\mathbf{h}$, resulting in the augmented vector

$$\mathbf{h}_t^{\text{aug}} = [\mathbf{h}; \mathbf{h}_t^{\text{noisy}}]. \tag{18}$$

Finally, $\mathbf{h}_{aug}$ is passed through a multi-layer perceptron (MLP) and a new classification head to generate the probability distribution for the next token:

$$p(\mathbf{y} \mid \mathbf{h}_t^{\text{aug}}) = \text{softmax}(\text{LM\_HEAD}((\text{MLP}(\mathbf{h}_{\text{aug}})))). \tag{19}$$

It is worth noting that for each batch, In addition to aligning $p(\mathbf{y} \mid \mathbf{h}_t^{\text{aug}})$ with the next token ground truth by minimizing the cross-entropy loss, we randomly generate two time steps, $t_1$ and $t_2$, and obtain logits $\text{logits}(\mathbf{y} \mid \mathbf{h}_{t_1}^{\text{aug}})$ and $\text{logits}(\mathbf{y} \mid \mathbf{h}_{t_2}^{\text{aug}})$. We then minimize the mean squared error (MSE) loss between them to ensure as much consistency as possible.

**Generation Phase** Now we describe the generation process. After the input passes through the decoder layers and obtains the last hidden states, we encounter a challenge during the inference phase since the next token $\hat{y}$ is unknown. To address this, we input a complete Gaussian noise vector $\mathbf{h}_{1000}^{\text{noisy}}$(i.e. $\epsilon$), which is denoised by model to generate the next token $\mathbf{y}_1$. This process is then iterated, with $\mathbf{y}_1$ serving as $\hat{y}$ for the subsequent iteration. Following the steps outlined in Eq. 16, 17, 18, and 19, we generate $\mathbf{y}_2$, and so on, iteratively.

The noise addition time step for each iteration is predetermined as a hyperparameter. In our experiments, we employ a linearly decreasing schedule for the time steps. For instance, with a maximum noise step of 1000 and 5 iterations, the time steps $t$ are set as $[1000, 800, 600, 400, 200]$, ensuring a gradual reduction of noise over the course of iterations.

## C. Supplementary Experimental Results

### C.1. Supplementary Comparison to Label Smoothing

We conducted experiments on the N-body simulation task (3 isolated particles, 2 sticks, and 1 hinge) using GCN as the backbone model. For label smoothing with continuous outputs, we implemented Gaussian noise injection via reparameterization: $\hat{y} = y + \mathcal{N}(0; \beta\mathbf{I})$ where $\beta$ controls the noise intensity and we use $\hat{y}$ to calculate loss. This optimization is verified to induce better representations [1]. Then, we further fine-tune the trained model on exact labels to ensure precise regression. The Prediction error ($\times 10^{-2}$) results of SL, PCL, and SL_label_smoothing with various $\beta$ are shown in Table 5. Experimental results demonstrate that while label smoothing achieves moderate improvements (best $\beta = 0.3$, 3.256 error) over standard supervised learning (SL, 3.453), PCL (2.795 error) maintains a significant performance advantage over even the optimal label smoothing configuration.

*Table 5.* Prediction error ($\times 10^{-2}$) of GCN model on N-body simulation task with 3 isolated particles, 2 sticks and 1 hinges. The comparison includes SL, PCL, and SL with label smoothing with varying $\beta$ values for smoothing intensity.

| Method | SL | PCL | $\beta = 0.01$ | $\beta = 0.03$ | $\beta = 0.05$ | $\beta = 0.1$ | $\beta = 0.3$ | $\beta = 0.5$ | $\beta = 1.0$ |
|---|---|---|---|---|---|---|---|---|---|
| Prediction Error | 3.453 | 2.795 | 3.460 | 3.461 | 3.428 | 3.418 | 3.256 | 3.258 | 3.368 |

## C.2. Supplementary Comparison to Ensemble Methods

Given that PCL requires multiple forward passes for inference, we supplement the comparison to ensemble methods. We conducted experiments on the N-body simulation task (3 isolated particles, 2 sticks, and 1 hinge) using GCN as the backbone model, comparing PCL against a bagging ensemble approach where n independent models were trained on the full dataset and their predictions averaged. The prediction error ($\times 10^{-2}$) results under matched inference passes (1-5) are presented in Table 6.

*Table 6.* Prediction error ($\times 10^{-2}$) of GCN model on N-body simulation task with 3 isolated particles, 2 sticks and 1 hinges. The comparison includes PCL and SL with the bagging technique under varying numbers of inference model passes.

| # Inference Passes | 1 | 2 | 3 | 4 | 5 |
|---|---|---|---|---|---|
| PCL | 2.834 | 2.818 | 2.808 | 2.796 | 2.786 |
| SL Bagging | 3.453 | 3.210 | 3.132 | 3.084 | 3.065 |

## C.3. Unconstrained N-body Simulation

The N-body simulation task involves predicting the future positions of a set of interacting particles over time based on initial conditions such as their positions, velocities, and the inherent physical forces governing their interactions (Satorras et al., 2021). The task's outputs retain the same dimensionality and complexity as the inputs. The evolution of particle positions and velocities follows fundamental physical laws such as gravitational or electrostatic interactions. We follow (Satorras et al., 2021) to solve the 5-charged-particle system in 3-dimensional space. The system consists of five particles, each with either a positive or negative charge, and each particle has an associated position and velocity.

**Dataset.** We collected 3000 trajectories for training, 2000 for validation, and 2000 for testing. Each trajectory spans 1000 timesteps. For each trajectory, the initial conditions include the particle positions $p(0) = \{p_1(0), \ldots, p_5(0)\} \in \mathbb{R}^{5 \times 3}$, the initial velocities $v(0) = \{v_1(0), \ldots, v_5(0)\} \in \mathbb{R}^{5 \times 3}$, and the respective charges $c = \{c_1, \ldots, c_5\} \in \{-1, 1\}^5$. The task is to predict the positions of the five particles after 1000 timesteps. The model is optimized by minimizing the averaged Mean Squared Error (MSE) between the predicted positions and the ground truth positions.

**Metrics.** We adopt two evaluation metrics to evaluate the regression quality for test data: 1) Mean Square Error (MSE): the average of the squares of the errors between the predicted values and the true values; 2) Mean Absolute Error (MAE): the average of the absolute differences.

**Model Design.** We consider the state-of-the-art graph modeling solution for this task, where we input the concatenation of the initial positions and the velocities as the node features. The charges are input as edge attributes $a_{ij} = c_i c_j$. We take the

*Table 7.* Comparison of traditional supervised learning and PCL for MSE.

| Method | MSE↓ | MAE↓ |
|---|---|---|
| GCN (Kipf & Welling, 2016) | 0.01064±0.00014 | 0.04322±0.00082 |
| GCN-PCL (Ours) | **0.00927±0.00020** | **0.03783±0.00018** |
| GAT (Veličković et al., 2017) | 0.00969±0.00040 | 0.03996±0.00198 |
| GAT-PCL (Ours) | **0.00910±0.00038** | **0.03726±0.00068** |
| GGNN (Li et al., 2015) | 0.01220±0.00020 | 0.04614±0.00146 |
| GGNN-PCL (Ours) | **0.01143±0.00042** | **0.04336±0.00127** |

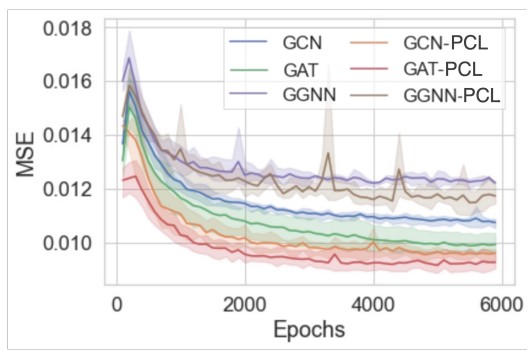

*Figure 6.* MSE curves on test data.

*Table 8.* Ablation study on loss construction.

| Method | MSE↓ | MAE↓ |
|---|---|---|
| Traditional SL | 0.01064 | 0.04322 |
| w/o $\lambda_1$-term, w/ $\lambda_2$-term | 4.34559 | 1.62437 |
| w/ $\lambda_1$-term, w/o $\lambda_2$-term | 0.00956 | 0.03895 |
| w/ $\lambda_1$-term, w/ $\lambda_2$-term | 0.00927 | 0.03783 |

model outputs as the estimated positions. To introduce PCL, we adopt two linear layers to encode the input attributes and the noised label, respectively, and then concatenate them to form the input hidden feature to the subsequent graph neural layers. We adopt 4 graph neural layers, and for each layer's output, we integrate the timestep embedding extracted by the sinusoidal position embedding and a linear layer through addition. In this task, $\mathbf{y} \in \mathbb{R}^{5 \times 3}$ and we adopt the Gaussian noising process to produce noised labels as shown in Eq. 7.

**Results.** We compare the model with the classic graph neural networks, including Graph Convolutional Networks (GCN) (Kipf & Welling, 2016), Graph Attention Network (GAT) (Veličković et al., 2017), and Gated Graph Neural Networks (GGNN) (Li et al., 2015). For each model, we compare the performance with the models trained by the classic SL and our proposed PCL. Table 7 shows the superiority of PCL on quantitative results with lower estimation errors on both MSE and MAE under same settings, and Fig. 6 shows performance gain on the test MSE curves within the training process. Table 8 provides ablation studies on the effects of the $\lambda_1$-term and $\lambda_2$-term and verifies the effectiveness of every loss term in Eq. 8.

### C.4. Supplementary Results for Semantic Segmentation

For semantic segmentation, to vividly illustrate the effects of PCL, we present a comparison of the Intersection over Union (IoU) metrics for all three models across the ADE20K dataset's 150 categories in Table 9. The results indicate that MobileNetV2dilated, ResNet50dilated, and HRNetV2 have achieved advantages in IoU on 70.00%, 58.67%, and 64.67% of the categories respectively after using PCL.

Fig. 7 illustrates the differences in predicting semantic segmentation maps among various models and training methods, visually reflecting the performance differences between the models as outlined in Table 3.

*Table 9.* Comparison of Intersection over Union (IoU) for classic supervised learning (SL) versus the proposed predictive consistency learning (PCL) in semantic segmentation across various neural backbones. Bold indicates better performance in that category. MoibleNet: MobileNetV2dilated, ResNet50: ResNet50dilated. Order: Ranked from top to bottom based on the probability of each category in the ADE20K dataset

| Object | MobileNet | | ResNet50 | | Object | MobileNet | | ResNet50 | | Object | MobileNet | | ResNet50 | |
|---|---|---|---|---|---|---|---|---|---|---|---|---|---|---|
| | SL | PCL | SL | PCL | | SL | PCL | SL | PCL | | SL | PCL | SL | PCL |
| wall | **69.01** | 67.29 | 73.21 | **77.57** | building | 75.72 | **77.00** | 80.89 | **85.21** | building | 92.78 | **93.14** | **93.50** | 93.36 |
| floor | **71.95** | 71.85 | 77.43 | **79.61** | tree | 68.86 | **69.99** | 70.12 | **72.07** | tree | 77.41 | **78.59** | 80.22 | **84.16** |
| road | **75.96** | 75.48 | 80.24 | **80.98** | bed | **80.13** | 80.04 | 84.53 | **86.09** | bed | **53.45** | 53.41 | 57.96 | **64.24** |
| grass | 64.91 | **67.28** | 66.54 | **75.82** | cabinet | 49.12 | **50.57** | 56.45 | **59.47** | cabinet | 54.10 | **54.77** | 60.93 | **63.53** |
| person | **70.96** | 70.75 | **75.13** | 72.22 | earth | 30.30 | **31.84** | 32.37 | **44.69** | earth | 32.27 | **32.86** | 39.90 | **52.50** |
| table | 45.08 | **46.04** | 52.93 | **57.66** | mount | 48.37 | **52.20** | 54.45 | **55.61** | mount | 45.41 | **47.46** | 45.40 | **55.84** |
| curtain | 61.68 | **63.74** | 65.41 | **71.90** | chair | 45.15 | **46.72** | 53.01 | **55.42** | chair | **76.57** | 75.96 | **81.05** | 79.83 |
| water | 49.54 | **51.51** | 52.56 | **60.90** | picture | 63.22 | **64.17** | 66.01 | **70.79** | picture | 55.32 | **56.09** | 59.25 | **67.82** |
| shelf | 34.45 | **35.85** | 40.74 | **48.58** | house | 26.73 | **37.92** | 54.76 | **59.03** | house | 51.19 | **55.54** | **54.69** | 17.62 |
| mirror | 42.59 | **43.43** | 54.74 | **59.80** | rug | 40.69 | **44.73** | 53.59 | **61.40** | rug | 20.00 | **23.24** | 24.74 | **46.71** |
| armchair | **33.24** | 33.23 | 38.42 | **49.39** | seat | 37.37 | **42.45** | **52.93** | 50.21 | seat | 27.98 | **29.42** | 31.38 | **48.18** |
| desk | 28.57 | **29.67** | **47.64** | 41.48 | rock | 31.21 | **33.04** | 35.79 | **50.45** | rock | 35.10 | **38.37** | 39.80 | **45.83** |
| lamp | **49.66** | 49.48 | 58.79 | **63.01** | bath | 60.63 | **64.66** | 69.45 | **70.42** | bath | 27.08 | **29.41** | 32.58 | **37.84** |
| cushion | 38.03 | **39.14** | **47.27** | 43.30 | base | 10.83 | **13.20** | 26.47 | **35.80** | base | 13.79 | **15.00** | 15.39 | **20.79** |
| column | 32.52 | **33.72** | 40.73 | **49.83** | signboard | 26.02 | **26.75** | 30.75 | **38.02** | signboard | 34.45 | **36.27** | 37.04 | **54.18** |
| counter | 17.63 | **21.31** | 24.55 | **51.29** | sand | 17.37 | **22.90** | 35.48 | **45.76** | sand | **50.58** | 50.17 | **65.30** | 59.64 |
| skyscraper | 41.18 | **45.66** | 53.45 | **67.52** | hearth | **63.57** | 63.18 | 62.90 | **70.00** | hearth | 51.17 | **52.91** | 68.53 | **70.14** |
| grandstand | 37.60 | **47.05** | **39.08** | 33.67 | path | **15.70** | 14.58 | 19.93 | **23.39** | path | 26.81 | **28.95** | 18.78 | **44.63** |
| runway | 49.63 | **50.55** | 57.36 | **60.86** | case | 34.37 | **39.37** | 44.64 | **54.33** | case | 88.14 | **88.33** | **89.86** | 85.98 |
| pillow | **42.76** | 41.31 | **46.58** | 37.59 | screen door | 45.60 | **48.47** | 58.18 | **63.30** | screen door | 28.34 | **29.06** | 19.93 | **48.86** |
| river | 12.43 | **12.92** | 11.70 | **27.58** | bridge | 32.22 | **36.03** | **55.18** | 37.89 | bridge | 29.15 | **31.21** | 37.23 | **39.30** |
| blind | 13.24 | **26.42** | 39.79 | **54.50** | coffee table | **48.14** | 43.74 | 57.95 | **63.86** | coffee table | 73.80 | **74.33** | **84.20** | 77.64 |
| flower | 26.52 | **30.22** | 33.88 | **42.41** | book | 36.39 | **39.23** | 41.95 | **44.01** | book | 5.31 | **5.82** | 5.35 | **27.00** |
| bench | 35.06 | **36.14** | 38.70 | **42.32** | countertop | 35.03 | **38.29** | **49.59** | 48.98 | countertop | 55.27 | **55.90** | **76.17** | 65.01 |
| palm | 36.59 | **38.48** | **47.14** | 45.50 | kitchen island | 25.78 | **27.30** | **36.10** | 23.53 | kitchen island | 47.56 | **48.16** | **53.04** | 46.73 |
| swivel chair | **35.47** | 33.70 | **47.66** | 41.91 | boat | 26.55 | **29.40** | 53.28 | **59.06** | boat | 21.39 | **29.08** | 23.88 | **47.72** |
| arcade machine | 22.63 | **32.14** | **47.78** | 47.26 | hovel | 25.65 | **26.17** | 12.55 | **15.46** | hovel | 57.20 | **62.79** | **76.84** | 70.25 |
| towel | 31.85 | **34.53** | **53.05** | 52.35 | light | 32.29 | **34.86** | 43.61 | **45.28** | light | 3.52 | **5.69** | 23.40 | **24.88** |
| tower | **26.91** | 25.90 | 38.69 | **51.83** | chandelier | 57.94 | **59.60** | 59.92 | **62.95** | chandelier | **8.82** | 5.80 | 21.74 | **42.11** |
| streetlight | **7.74** | 7.19 | **22.78** | 22.20 | booth | 26.28 | **32.90** | 41.30 | **57.26** | booth | 45.35 | **49.25** | 63.16 | **68.86** |
| airplane | **39.72** | 39.50 | 50.81 | **54.55** | dirt track | **11.28** | 10.38 | 6.97 | **22.89** | dirt track | **24.60** | 23.67 | 28.22 | **39.82** |
| pole | **12.51** | 12.20 | 13.94 | **16.40** | land | **0.94** | 0.69 | 0.46 | **18.23** | land | 1.44 | **2.17** | 9.50 | **24.90** |
| escalator | 5.27 | **7.88** | 21.89 | **42.77** | ottoman | **28.07** | 26.99 | **38.95** | 38.36 | ottoman | 8.29 | **13.26** | **34.92** | 10.97 |
| buffet | **33.48** | 33.36 | 37.03 | **49.54** | poster | **10.05** | 7.30 | 20.61 | **37.01** | poster | 9.03 | **9.04** | 5.45 | **24.44** |
| van | **25.37** | 23.07 | 41.20 | **45.06** | ship | 1.72 | **26.34** | 24.27 | **83.96** | ship | 7.06 | **15.23** | 16.68 | **71.20** |
| conveyor | 46.77 | **46.93** | 38.53 | **44.23** | canopy | 9.79 | **14.49** | 15.35 | **29.41** | canopy | 53.79 | **59.10** | 59.56 | **76.23** |
| plaything | 7.95 | **13.39** | **20.22** | 16.25 | natatorium | 22.26 | **25.75** | **24.61** | 21.81 | natatorium | 26.78 | **27.75** | **37.90** | 36.75 |
| barrel | **6.54** | 6.47 | **23.75** | 22.49 | basket | 15.64 | **16.77** | **27.50** | 27.00 | basket | 35.83 | **38.82** | **49.53** | 35.14 |
| tent | 76.36 | **82.72** | 77.62 | **84.89** | bag | 1.93 | **4.91** | 9.18 | **16.85** | bag | 35.53 | **35.81** | **49.52** | 40.93 |
| cradle | **66.27** | 64.05 | **74.45** | 65.07 | oven | **33.26** | 30.10 | 35.46 | **44.45** | oven | 29.42 | **30.76** | 30.93 | **32.86** |
| food | 33.55 | **45.09** | **43.66** | 40.65 | step | 1.58 | **3.30** | 0.14 | **17.22** | step | 12.07 | **27.30** | 22.64 | **44.84** |
| brand | 18.18 | **20.85** | 20.95 | **32.78** | microwave | **33.01** | 32.82 | **34.16** | 27.20 | microwave | 26.52 | **28.90** | 39.99 | **51.19** |
| animal | 42.52 | **43.54** | 54.43 | **56.01** | bicycle | **31.74** | 31.13 | **47.76** | 39.18 | bicycle | 3.41 | **12.23** | **37.02** | 1.47 |
| dishwasher | 44.65 | **45.73** | **66.85** | 44.28 | screen | 56.52 | **57.84** | 61.08 | **68.69** | screen | 0.30 | **1.74** | 5.77 | **29.02** |
| sculpture | 2.79 | **9.56** | 18.52 | **29.25** | hood | 31.01 | **35.97** | **49.64** | 34.65 | hood | **22.72** | 21.80 | **38.07** | 36.85 |
| vase | **18.52** | 16.69 | **36.89** | 32.72 | stoplight | **5.44** | 1.98 | **26.40** | 18.00 | stoplight | 0.83 | **0.93** | 3.42 | **6.23** |
| ashcan | 20.05 | **25.89** | **39.94** | 35.85 | fan | **39.64** | 33.16 | 52.23 | **56.63** | fan | 23.69 | **28.73** | **31.10** | 17.93 |
| crt screen | 0.75 | **1.15** | 0.70 | **17.78** | plate | **33.26** | 31.50 | **42.14** | 41.12 | plate | 5.91 | **21.66** | 1.96 | **58.89** |
| notice board | 18.69 | **23.73** | **37.39** | 23.26 | shower | 0.00 | **0.01** | 0.00 | **0.20** | shower | 26.37 | **28.49** | 47.29 | **60.18** |
| glass | 2.08 | **3.87** | **10.87** | 8.51 | clock | 4.92 | **5.91** | **25.50** | 21.65 | clock | 9.82 | **10.00** | 21.03 | **41.11** |

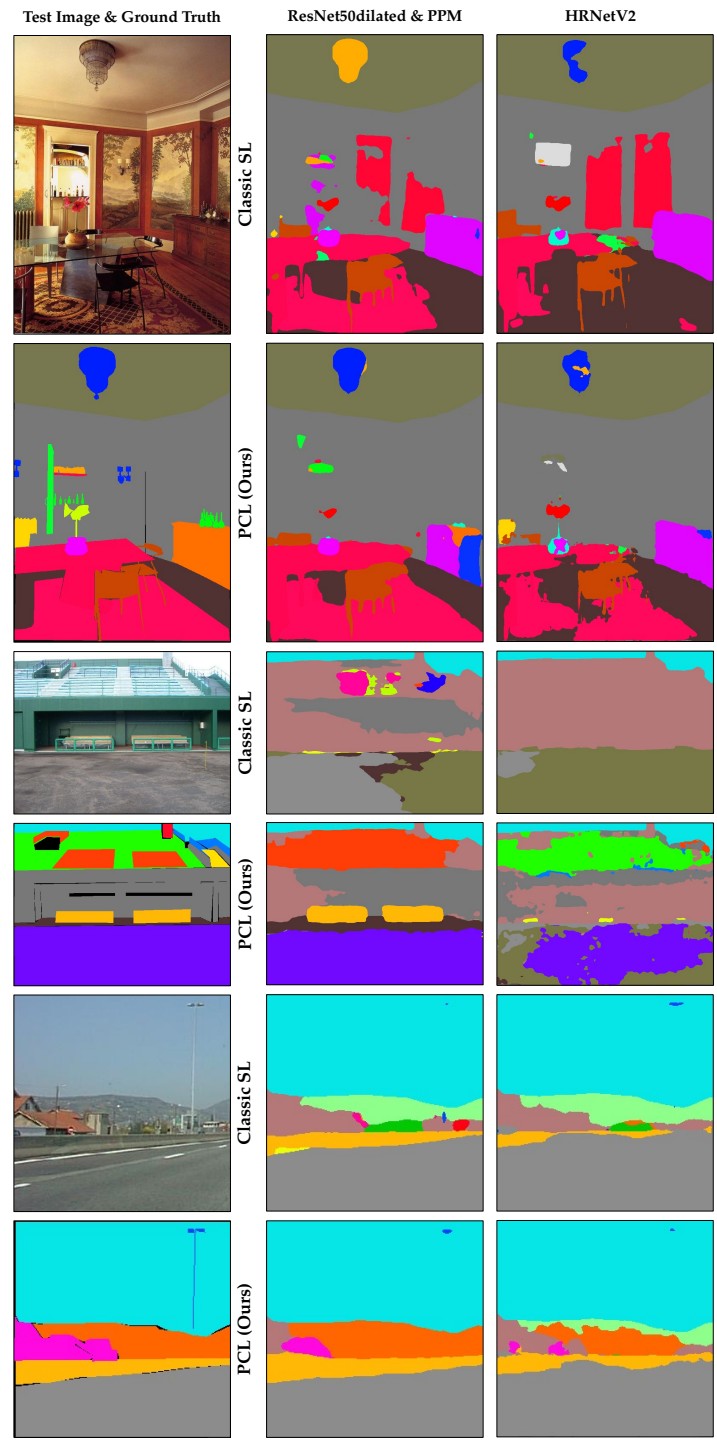

*Figure 7.* Comparison of classic supervised learning and PCL on semantic segmentation across different neural backbones.

