# OpenReview forum: "Generative Modeling Reinvents Supervised Learning: Label Repurposing with Predictive Consistency Learning"
_ICML.cc/2025/Conference — ICML 2025 poster_

### Official Review · Reviewer_yH7y · 2025-03-14

**Overall Recommendation:** 3

**Summary:**

Traditional methods predict labels directly from data inputs, but recent tasks often involve more complex labels. To address this, the authors propose Predictive Consistency Learning (PCL), inspired by consistency training in generative models. Unlike traditional approaches, PCL incorporates noise-perturbed labels as additional references, ensuring consistency across varying noise levels. It learns the relationship between latent features and a range of label information, enabling progressive learning and multi-step inference akin to gradual denoising, thus improving prediction quality. Experiments on vision, text, and graph tasks demonstrate the superiority of PCL.

**Claims And Evidence:**

The authors have demonstrated the effectiveness of their method by testing it across a variety of settings, including n-body simulation, semantic segmentation, and next-token prediction. These experiments highlight the versatility of PCL.

**Essential References Not Discussed:**

NA

**Experimental Designs Or Analyses:**

The proposed multi-step prediction approach brings up concerns regarding its efficiency. How much longer does the training process take compared to standard supervised learning? Additionally, what about the inference time? The authors may provide this information.

**Methods And Evaluation Criteria:**

The proposed methods and evaluation criteria generally make sense.

**Other Comments Or Suggestions:**

NA

**Other Strengths And Weaknesses:**

NA

**Questions For Authors:**

Given that it requires multiple forward passes for inference, how does its performance compare to that of ensemble methods?

**Relation To Broader Scientific Literature:**

Label smoothing involves applying some form of “perturbation” to the labels to help the model generalize better. How much of a performance advantage does PCL offer over label smoothing? The authors may report the results on a small dataset.

[1] Rethinking the Inception Architecture for Computer Vision. CVPR 2016

**Theoretical Claims:**

This paper does not make theoretical claims.

---

> ### Author Rebuttal · Authors · 2025-03-31
>
> Thanks for the valuable comments, nice suggestions, and for acknowledging our work. Below we respond to your specific comments.
>
> > **Q1: The proposed multi-step prediction approach brings up concerns regarding its efficiency. How much longer does the training process take compared to standard supervised learning? Additionally, what about the inference time?**
>
> For training, since the loss calculation requires two inference predictions with different noise levels during training, it requires twice the training cost of the traditional supervised learning paradigm. Moreover, it typically takes less than twice the iteration for the model to converge.
>
> For inference, PCL can produce much superior predictions with merely a single forward pass as SL does. Meanwhile, PCL can achieve better performance with more forward passes. As shown in Fig. 3 using a GCN backbone on the 3,2,1 dataset, even single-step PCL inference achieves superior performance (prediction error 0.02835) compared to SL (0.03478), while additional inference steps (e.g., 5-step) can further refine results (0.02795).
>
> > **Q2: Label smoothing involves applying some form of “perturbation” to the labels to help the model generalize better. How much of a performance advantage does PCL offer over label smoothing? The authors may report the results on a small dataset.**
>
> Thanks for this valuable point. To evaluate this, we conducted experiments on the N-body simulation task (3 isolated particles, 2 sticks, and 1 hinge) using GCN as the backbone model.
>
> For label smoothing with continuous outputs, we implemented Gaussian noise injection via reparameterization: $\hat{\mathbf{y}} = \mathbf{y} + \mathcal{N}(0;\beta\mathbf{I})$ where $\beta$ controls the noise intensity and we use $\hat{\mathbf{y}}$ to calculate loss. This optimization are verified to induce better representations [1]. Then, we further finetune the trained model on exact labels to ensure precise regression. The Prediction error ($\times 10^{-2}$) results of SL, PCL, and SL_label_smoothing with various 𝛽 are as follows:
>
>
>
> | |SL| PCL| 𝛽=0.01 | 𝛽=0.03 | 𝛽=0.05 | 𝛽=0.1 | 𝛽=0.3 | 𝛽=0.5 | 𝛽=1.0 |
> | -| -| -| - | -| -| -| - | -| -|
> | Prediction Error | 3.453 | 2.795 | 3.460  | 3.461  | 3.428  | 3.418 | 3.256 | 3.258 | 3.368 |
>
> The results show that while label smoothing yields certain improvements over standard supervised learning (with the best performance at β = 0.3), it still falls short of PCL by a significant margin. Specifically, PCL achieves a prediction error of 2.795, clearly outperforming both SL and the best-performing label smoothing setup (3.256). This highlights PCL’s stronger ability to generalize through its learned iterative refinement process.
>
>
> [1] How Does a Neural Network’s Architecture Impact Its Robustness to Noisy Labels? NeurIPS 2021.
>
>
> > **Q3: Given that it requires multiple forward passes for inference, how does its performance compare to that of ensemble methods?**
>
>
> Thank you for the insightful question. To evaluate this comparison, we conduct experiments on the N-body simulation task (comprising 3 isolated particles, 2 sticks, and 1 hinge), using a GCN backbone. We compare PCL to a standard bagging ensemble approach, where $n$ independent models are trained on the full dataset. At inference time, predictions from the $n$ models are averaged to reduce individual model biases. The prediction error ($\times 10^{-2}$) across different numbers of inference passes (i.e., number of independent models or PCL steps) is summarized below:
>
>
> | #inference passes | 1 | 2| 3| 4| 5|
> | -| -| -| -| -| -|
> | PCL| 2.834 | 2.818 | 2.808 | 2.796 | 2.786 |
> | SL (Bagging) | 3.453 | 3.210 | 3.132 | 3.084 | 3.065 |
>
>
> As the results show, while bagging does provide performance improvements with more inference passes, it consistently underperforms compared to PCL. Notably, even with five inference passes, bagging does not reach the accuracy of PCL’s single-pass result (2.786 vs. 2.834), highlighting the efficiency and effectiveness of PCL’s learned iterative refinement.
>
> The computational trade-offs further emphasize PCL's advantages. Bagging incurs significant overhead—requiring training, storing, and maintaining $n$ separate models, resulting in O(n) training and memory costs. In contrast, PCL achieves its gains through a single model, keeping both training and deployment lightweight while delivering superior predictive performance.
>
>
> ---
>
> We hope this response could help address your concerns, and we are more than happy to address any further concerns you may have.

---

### Official Review · Reviewer_bsic · 2025-03-14

**Overall Recommendation:** 3

**Summary:**

This paper aims to leverage more rich label information to enhance supervised learning. To achieve this goal, it proposes a new learning method called PCL (predictive consistency learning). Different from those conventional supervised learning approaches, PCL learns from both the labels as well as the noise-perturbed label information. Besides, PCL also optimizes the consistency across different noise levels to ensure the learning quality. Experiments on multiple tasks demonstrate the effectiveness of the proposed method.

**Claims And Evidence:**

Weaknesses:
- There seems to be an overclaim regarding the label modeling. In supervised learning, the label tends to correspond to the annotations from crowd sourcing. The recovered noise from the label is also related to the recovering model, which is different from the label itself. Therefore, it is improper to claim that the label information is more complex than the inputs.

**Essential References Not Discussed:**

Weaknesses:
- It seems that the references are a bit out-of-date. Most references are from or earlier than 2023. Is there any more recent related work?

**Experimental Designs Or Analyses:**

Strengths:
- The experiments on multiple datasets demonstrate the effectiveness of the proposed method, as the performance gain is satisfactory.

Weaknesses:
- The ablation studies are only conducted on the constrained N-body simulation task. There is no ablation studies for other tasks (semantic segmentation and supervised next-token prediction fine-tuning).
- It seems that the authors have not reported the hyperparameter settings. Besides, there is no hyperparameter sensitivity analysis.

**Methods And Evaluation Criteria:**

Strengths:
- The proposed method has been adequately evaluated on multiple tasks. The authors have also provided thorough explanations for the experimental settings and criteria.

Weaknesses:
- The computational cost of the proposed method should be quite expensive, as it learns to model the noise using an additional model. Therefore, it is unfair to directly compare PCL with supervised learning. Moreover, it is necessary to analyze the complexity of the proposed method.

**Other Comments Or Suggestions:**

See above.

**Other Strengths And Weaknesses:**

See above.

**Questions For Authors:**

See above.

**Relation To Broader Scientific Literature:**

Strengths:
- This paper is related to supervised learning and generative models such as Diffusion. The authors have properly discussed them.

**Theoretical Claims:**

Weaknesses:
- There is no theoretical evidence for supporting the proposed PCL.

---

> ### Author Rebuttal · Authors · 2025-04-01
>
> We sincerely appreciate the valuable comments and nice suggestions. However, we believe there may exist some misunderstandings regarding the methodology. We regret any confusion caused by our presentation and would like to clarify the core methodology of PCL.
>
> What PCL does is to leverage the concept of consistency modeling to enhance the traditional predictive paradigms (e.g. in supervised learning), which modifies the mapping manner in training from `data input x -> full label y` to  `data input x, noised label y_t (as additional hints) -> full label y`, and then enforce mapping consistency across different noise levels (controlled by t) to regulate label information content during training. The "label modeling" in the title refers to systematically modulating label information density (which is achieved by gradually adding noise to the raw labels and providing noised labels as input to make the model learn the complementary label information) and PCL does not employ an auxiliary model to predict or recover noise but rather introduces a novel training paradigm.
>
> > **Q1: There seems to be an overclaim regarding the label modeling. It is improper to claim that the label information is more complex than the inputs.**
>
> To clarify, we have not claimed that label information is universally more complex than inputs in the paper. Rather, we highlight (e.g., in L30 right) that "recent advanced scenarios involve much more complex labels" and "these challenges expose predictive bottlenecks due to the inherent complexity of transformations from features to labels, in addition to feature extraction". The label complexity is for comparison to simple categorical annotation tasks, rather than compared to input complexity.
>
> In many modern tasks, particularly those beyond traditional classification, labels and inputs can share closer dimensionality and information density, representing a significant departure from simple categorical annotation tasks. For instance, in LLM fine-tuning, both questions (inputs) and answers (labels) are natural language sequences requiring similar levels of semantic understanding and generation capability.
>
>
> > **Q2: The computational cost. PCL learns to model the noise using an additional model.**
>
> PCL is within the supervised learning paradigm and **does not employ an auxiliary model** to predict or recover noise. From the model perspective, it directly predicts the labels just as SL does. The mere difference lies in that it receives additional label hints as input. Thus, the additional computational overhead is minor in a forward pass.
>
> During training, since the loss calculation requires two inference predictions with different noise levels, it requires twice the training cost of the traditional supervised learning paradigm. Moreover, it typically takes less than twice the iteration for the model to converge. For inference, PCL can produce much superior predictions with merely a single forward pass as SL does. More forward passes can lead to further performance gains. As shown in Fig. 3 using a GCN backbone on the 3,2,1 dataset, even single-step PCL inference achieves superior performance (prediction error 0.02835) compared to SL (0.03478), while additional inference steps (e.g., 5-step) can further refine results (0.02795).
>
> > **Q3: The ablation studies are only conducted on the N-body simulation task. No ablation studies for other tasks.**
>
> The purpose of our experiments across three modalities is to validate the broad applicability of our method. We chose to conduct a more detailed analysis on one task to ensure the depth of our analysis.
>
> Here we supplement the ablation study of $\lambda_2$ term on the semantic segmentation task in [table](https://anonymous.4open.science/r/PCL-0F54/lambda2-segmentation.png) and we find excluding excluding the $\lambda_1$ term causes the model to fail.
>
> > **Q4: Hyperparameter settings and sensitivity analysis.**
>
> The key model-specific hyperparameters are set as $\alpha=0.2$ and $\lambda_1=\lambda_2=1$ in our experiments, with $\lambda$ values chosen for equal weighting rather than through extensive tuning. In response to your valuable suggestion, we conduct sensitivity analyses across reasonable parameter variations. All other parameters follow standard SL configurations as detailed in the paper, with all source code to be fully disclosed upon acceptance.
>
> The experimental results for hyperparameters:
>
> $\alpha$: [Table](https://anonymous.4open.science/r/PCL-0F54/alpha.png)
>
> $\lambda_1$: [Table](https://anonymous.4open.science/r/PCL-0F54/lambda1.png)
>
> $\lambda_2$: [Table](https://anonymous.4open.science/r/PCL-0F54/lambda2.png).
>
> > **Q5: Source code.**
>
> The open-source release of PCL is crucial for its impact on the research community, and we commit to open-sourcing our code upon acceptance of the final manuscript.
>
> ---
> We hope this response could help address your concerns, and we are more than happy to address any further concerns you may have.

---

> > ### Comment · Reviewer_bsic · 2025-04-05
> >
> > Thank you for your valuable explanations, which have addressed my misunderstandings and concerns. After reading the rebuttal and the comments from other reviewers, I would like to raise my score to 3.

---

> > > ### Author Response · Authors · 2025-04-07
> > >
> > > Thank you for your thoughtful feedback and for acknowledging our efforts in addressing the initial concerns. We sincerely appreciate your constructive engagement throughout this process and the valuable suggestions you have provided. We will carefully incorporate the additional explanations into the manuscript to enhance clarity and ensure that the key points are more clearly articulated.
> > >
> > > As the discussion period remains open, we warmly welcome any further feedback from all the reviewers to help us refine the paper further. Thank you again for your time and expertise.

---

### Official Review · Reviewer_FjUw · 2025-03-14

**Overall Recommendation:** 4

**Summary:**

This paper introduces a new prediction paradigm for more complex label spaces than the ones that are traditionally assumed in supervised learning. The paper draws inspiration from the "generative consistency learning" learning paradigm to produce the predictive consistency learning (PCL) paradigm, which the authors suggest is more suited to complex label spaces. The technique involves predicting the label with progressive amounts of label noise so that models can make predictions even when the ground truth labels contain varying amounts of noise, and enables multi-step denoising techniques within the PCL framework. Experimentally, the authors demonstrate PCL compared to supervised learning in a variety of settings: N-body simulation, semantic segmentation, and language modeling.

**Claims And Evidence:**

The main claim of the paper is that the proposed PCL is better suited to complex label spaces compared to supervised learning. The experimental evidence for this suggests that this is true -- in all of the settings that the authors evaluated, N-body simulation, semantic segmentation, and language modeling, PCL indeed outperforms supervised learning, and these settings all correspond to what I would think of as having more complex label spaces.

**Essential References Not Discussed:**

From what I can tell, the essential references were mostly covered. If anything, the related work seems to be missing a discussion of how this work relates to the field of structured prediction, as such label spaces seem to be a key motivating component of this work.

**Experimental Designs Or Analyses:**

The experimental design seemed sound from my read-through. The N-body simulation is a graph learning problem, so the authors use appropriate supervised graph learning baselines. Similarly, the baselines were chosen appropriately for the semantic segmentation and language modeling tasks.

**Methods And Evaluation Criteria:**

The evaluation protocol makes sense to me. N-body simulations, semantic segmentation, and language modeling are all tasks which involve complex label spaces that are typically modeled using supervised learning. The proposed method is described clearly, although in the abstract, it was initially unclear what generative consistency models refer to -- more exposition in the abstract would help for readers who are unfamiliar with this area.

**Other Comments Or Suggestions:**

On my first read, I found the abstract to be a somewhat confusing read. "Directly predicting labels from data inputs has been a long-standing default mechanism for label learning tasks, e.g., supervised learning" seems like an unnecessarily roundabout way to describe supervised learning -- are there other examples of label learning tasks beyond supervised learning and this work? This statement seems to imply this. Later on in the abstract, the authors mention the "generative consistency training concept in generative consistency models," which I was unfamiliar with -- this part should have somewhat more exposition.

**Other Strengths And Weaknesses:**

**Strengths**
- The proposed PCL is a novel and interesting take on the problem of learning labels from inputs, and tackles the relevant challenge of learning from more complex and structured labels.
- The proposed framework is described in a clear and easy-to-follow way.
- The experimental setup seems to appropriately cover complex label spaces including graphs, pixel-level segmentation, and language modeling.

**Weaknesses**
- I might have misunderstood, but it was unclear to me how the choice of noise distribution would impact results. Does the noise distribution have to be similar to the natural label noise distribution?
- Again I might have missed this, but I can't tell what noise distribution was used for the N-body simulation problem or the language modeling task.
- It's not immediately clear how "next-token prediction" is a complex label space -- if the label space is defined as predicting a single token from the vocabulary, then it seems to simply be a large label space. On the other hand, I would think that that the task of predicting entire sequences (all at once) would be a more obvious choice for a more complex structured label space, perhaps applied to a sequence to sequence learning problem.

**Questions For Authors:**

As mentioned before, my main question is about the noise distributions used in the experiments and how the choice of noise distribution impacts results. Furthermore, for complex label spaces, it might be unclear or challenging to design a realistic noise distribution -- how might such cases be handled?

**Relation To Broader Scientific Literature:**

The work is motivated by the need for learning paradigms targeting modern complex label spaces -- the current literature often comprises techniques that are largely based on traditional supervised learning, where complex or structured label spaces are perhaps handled in an ad hoc way.

**Theoretical Claims:**

I did not carefully verify the correctness of any of the math introduced in the paper, but it seemed to make sense from my initial read.

---

> ### Author Rebuttal · Authors · 2025-03-31
>
> Thanks for the valuable comments, nice suggestions, and for acknowledging our work. Below we respond to your specific comments.
>
> > **Q1: How this work relates to the field of structured prediction.**
>
> Thanks for your insightful suggestion. Indeed, our work also handles the challenge of structured prediction, i.e., learning to predict complex, structured outputs. However, we observe that with the widespread adoption of deep neural networks, many structured prediction problems (like sequence or graph prediction) can now be effectively handled through standard feed-forward architectures trained end-to-end, often being naturally subsumed under the broader umbrella of supervised learning. While classical structured prediction methods may focus on specific output structures (e.g., trees or sequences), our approach makes no assumptions about certain structures, while being particularly effective for handling complex labels. This is why we emphasize label complexity within the supervised learning paradigm rather than through the lens of structured prediction methods. We sincerely appreciate your suggestion and will expand our discussion of related work in this field during revision.
>
>
> > **Q2: Does the noise distribution have to be similar to the natural label noise distribution?**
>
> The noise distribution does not necessarily need to match the natural label noise distribution, as our primary goal in Sec. 4.2 is to design a noise process that systematically degrades label information in a controlled manner. The key is to tailor the noise to the structure of the label space (whether continuous or discrete) rather than replicating real-world noise patterns. For continuous labels, we employ Gaussian noise to gradually corrupt the signal, while for discrete labels, we use categorical noise that diffuses the probability mass of one-hot vectors across classes. In cases where the discrete label space is excessively large or complex, we shift to perturbing the latent representations directly with continuous noise. These approaches comprehensively cover the spectrum of label types (continuous or discrete), which is also why we choose these tasks in experiments, which can exemplify different scenarios.
>
> > **Q3: What noise distribution was used for the N-body simulation problem or the language modeling task?**
>
> For N-body simulation, the label $\mathbf{y}\in \mathbb{R}^{p\times 3}$ where $p$ denotes the number of particles. Since the labels maintain continuous values, we adopt Gaussion noise for the noising process: $q(\mathbf{y}_t | \mathbf{y}) = \mathcal{N}(\mathbf{y}_t; \sqrt{\bar{\alpha}_t} \mathbf{y}, (1-\bar{\alpha}_t)\mathbf{I})$ where $\bar{\alpha}_t$ controls the noise level.
>
> For language modeling, since the raw token space is extremely large, we directly add noise to the latent embeddings of the labels, which are multi-dimensional continuous vectors. we adopt Gaussion noise for the noising process: $q(\mathbf{h}\_{\mathbf{y}\_t} | \mathbf{h}\_{\mathbf{y}}) = \mathcal{N}(\mathbf{h}_{\mathbf{y}_t}; \sqrt{\bar{\alpha}\_t} \mathbf{h}\_{\mathbf{y}}, (1-\bar{\alpha}\_t)\mathbf{I})$.
>
> The details can be found in Sec. 4.2.
>
> > **Q4: It's not immediately clear how "next-token prediction" is a complex label space.**
>
> Next-token prediction, while seemingly maintaining simple labels, actually operates in a complex label space due to the vast vocabulary size (often tens or hundreds of thousands of tokens) and the semantic information encoded in each prediction. Though it only predicts one token at a time, the process implicitly models long-range dependencies and coherent reasoning, as each step conditions on the full context to generate meaningful continuations. The high dimensionality of the token embedding space (e.g., 4096-dimensional vectors), where we apply noise, also underscores its complexity. We chose this task for its practical relevance in modern LLMs and its ability to demonstrate our method’s scalability to large label spaces. Sequence-to-sequence tasks are indeed another compelling setting, and we appreciate the suggestion and will explore such extensions as a natural direction for future work.
>
> > **Q5: Are there other examples of label learning tasks beyond supervised learning and this work?**
>
> The reason we haven’t strictly confined PCL to supervised learning is that the mapping from X to Y is fundamental across many learning paradigms, including semi-supervised, weakly supervised, and even self-supervised settings. While our current experiments focus on standard supervised learning for clarity and validation, PCL’s core mechanism (refining the X-to-Y mapping) is intentionally designed to be adaptable. We believe this flexibility opens doors for future extensions into broader learning frameworks, and we’re excited to see explorations in these directions in subsequent works.
>
> ---
>
> We hope this response could help address your concerns, and we are more than happy to address any further concerns you may have.

---

> > ### Comment · Reviewer_FjUw · 2025-04-09
> >
> > Thank you for the clarifications, these are quite helpful! I will maintain my score and recommend acceptance.

---

> > > ### Author Response · Authors · 2025-04-09
> > >
> > > We are deeply grateful for your thoughtful feedback and continued support. Your constructive engagement and valuable suggestions have been instrumental in this process. We will carefully incorporate the additional explanations into the manuscript to enhance clarity and ensure that the key points are more clearly articulated. Thank you once more.

---

### Official Review · Reviewer_Fty8 · 2025-03-14

**Overall Recommendation:** 3

**Summary:**

The paper introduces Predictive Consistency Learning (PCL), a paradigm for tasks with complex labels. The paper starts with arguing that traditional supervised learning struggles with high-dimensional or structured labels due to the difficulty of mapping compressed input features directly to rich label spaces. Inspired from image diffusion, PCL addresses this by decomposing label learning into a progressive process: it introduces noise-perturbed labels as additional inputs during training and enforces consistency across predictions at different noise levels. Experiments on vision (semantic segmentation), graph (N-body simulation), and text (LLM fine-tuning) tasks demonstrate PCL's effectiveness over standard supervised learning.

**Claims And Evidence:**

The claims are supported by empirical evidence across diverse tasks.

**Essential References Not Discussed:**

No.

**Experimental Designs Or Analyses:**

Experiments are sound but can be made more through with comparison against advanced baselines such as diffusion models for structured labels (for e.g. SegDiff [1] for segmentation).

---

[1] SegDiff: Image Segmentation with Diffusion Probabilistic Models, https://arxiv.org/pdf/2112.00390

**Methods And Evaluation Criteria:**

Yes.

**Other Comments Or Suggestions:**

I think [extreme classification](http://manikvarma.org/downloads/XC/XMLRepository.html) can be a good benchmark for this work (as it inherently has a high dimensional predictive output space)

**Other Strengths And Weaknesses:**

### Strengths
- Paper is well presented with clear details
- Experiments encompass multiple modality benchmarks

### Weakness
- The thesis of the paper is that for complex output spaces standard supervised learning is not appropriate, it also acknowledges that there are methods which can project the output space to a simpler latent space but provides no qualitative comparison against them.

**Questions For Authors:**

1. In what *predictive* cases is learning an invertible function from $Y_E \rightarrow Y$ not straightforward (hence necessating the need of the proposed approach)? dual encoder style modeling (which has a very simple inverse map) should be possible for most predictive tasks, right?
2. How does PCL differ from curriculum learning strategies that gradually increase label complexity?
3. Is there a train / inference time difference in the SL vs PCL training / inference?

**Relation To Broader Scientific Literature:**

PCL builds on consistency models and diffusion-based training but adapts them to deterministic label prediction. It connects to curriculum learning (progressive label complexity) and residual learning (noise as input).

**Theoretical Claims:**

N/A

---

> ### Author Rebuttal · Authors · 2025-03-31
>
> Thanks for the valuable comments, nice suggestions, and for acknowledging our work. Below we respond to your major comments.
>
> > **Q1: Qualitative comparison to methods that project the output space to a simpler latent space.**
>
> Introducing an encoder-decoder structure with a  $Y\to Y_E$ encoder and a $Y_E\to Y$ decoder may help handle the complexity of $Y$. This approach is commonly used in generative models like Latent Diffusion Models (LDMs), where data (e.g., images or videos) are first compressed into a latent space for generation before being decoded back into the original domain. However, for predictive mapping from $X$ to $Y$, which is a fundamental component in ML, explicitly introducing an encoder-decoder stage adds computational burden. During inference, the encoder (mapping $Y \to Y_E$) is redundant, yet its inclusion increases training complexity. Additionally, the two-stage learning process (first $X \to Y_E$, then $Y_E \to Y$) can lead to error accumulation, degrading overall performance.
>
> In contrast, our method retains the simplicity of traditional supervised learning while enhancing performance through progressive label decomposition. This approach allows for more effective learning of label information with minimal yet refined modifications to the standard training framework.
>
>
> > **Q2: How does PCL differ from curriculum learning (CL) strategies that gradually increase label complexity?**
>
> The key differences between PCL and CL can be summarized as follows:
>
> 1. **Learning Target Consistency**. CL predefines staged learning targets (e.g., coarse-to-fine labels), where each stage learns an approximation of the true label. This risks error accumulation, i.e., biased features from early stages may propagate to later stages. In contrast, the objective of PCL remains stable: always predicting the true label, with complexity dynamically adjusted by the input noise level.
> 2. **Progressive vs. Simultaneous Learning**. CL follows a fixed, sequential progression (e.g., easy→hard labels). PCL, however, randomly samples time steps during training, enabling the model to learn various label information levels (from partial to complete) simultaneously, where the time step t acts as a controller for label granularity.
> 3. **Controllable Prediction**. PCL’s noise-conditioned framework allows explicit control over prediction granularity via time step t. As evidenced in Fig. 5, setting a larger t tends to encourage the model to improve broader structural relationships, while setting a smaller t encourages the model to focus on finer details. Further, PCL supports multi-step inference (Sec. 4.4), where predictions are iteratively refined (analogous to diffusion denoising), boosting final accuracy.
> 4. **Leveraging Partial Labels as Input**. One of the motivations of PCL is to treat labels not just as targets but as learning facilitators. By feeding partially noised labels (e.g., "hints") during training, the model learns to exploit intermediate information (similar to how humans use reference solutions to help solve problems).
>
>
> > **Q3: Is there a train / inference time difference in the SL vs PCL training / inference?**
>
> For training, since the loss calculation requires two inference predictions with different noise levels during training, it requires twice the training cost of the traditional supervised learning paradigm. Moreover, it typically takes less than twice the iterations for the model to converge.
>
> For inference, PCL can produce much superior predictions with merely a single forward pass as SL does. Meanwhile, PCL can achieve better performance with more forward passes. As shown in Fig. 3 using a GCN backbone on the 3,2,1 dataset, even single-step PCL inference achieves superior performance (prediction error 0.02835) compared to SL (0.03478), while additional inference steps (e.g., 5-step) can further refine results (0.02795).
>
> > **Q4: Experiments are sound but can be made more thorough with comparison against advanced baselines such as diffusion models for structured labels.**
>
> Thanks for the acknowledgment and suggestion. Our method primarily takes traditional supervised learning as our main framework, where PCL is constructed within. The generative modeling approach you mentioned essentially reformulates the prediction task as a generation task, representing one specific implementation under the broader supervised learning framework ($X\to Y$ mapping). PCL operates at a different conceptual level, whose methodology for label utilization could potentially be integrated into diffusion-based prediction as well. In a word, our current framework is a bit orthogonal to and can be combined with the diffusion models.
>
> In our revision, we will discuss relevant works in this direction and explore subsequent attempts in future work.
>
> ---
>
> We hope this response could help address your concerns, and we are more than happy to address any further concerns you may have.

---

### Decision · Program_Chairs · 2025-05-01

**Decision:**

Accept (poster)

**Comment:**

This paper draws inspiration from diffusion models and introduces a predictive consistency learning paradigm for tasks with complex labels. The prior approaches often suffer in cases with high-dimensional or structured labels. This paper introduces noise perturbation to labels as additional inputs and forces the training model to be consistent in its predictions at different noise levels. Experiments on a variety of tasks (with different modalities too) demonstrate the effectiveness of the proposed method. All reviewers agreed on the clarity of the paper and its technical contributions.